



# An analysis of instabilities and limit cycles in glacier-dammed reservoirs

Christian Schoof

Department of Earth, Ocean and Atmospheric Sciences, University of British Columbia, Vancouver, Canada

*Correspondence to:* Christian Schoof
(cschoof@eoas.ubc.ca)

**Abstract.** Glacier lake outburst floods are common glacial hazards around the world. How big such floods can become (either in terms of peak discharge or in terms of total volume released) depends on how they are initiated: what causes the runaway enlargement of a subglacial or other conduit to start, and how big can the lake get before that point is reached? Here we investigate how the spontaneous channelization of a linked-cavity drainage system controls the onset of floods. In agreement with previous work, we show that floods only occur in a band of water throughput rates, and identify stabilizing mechanisms that allow steady drainage of an ice-dammed reservoir. We also show how stable limit cycle solutions emerge from the instability, a show how and why the stability properties of a drainage system with spatially spread-out water storage differ from those where storage is localized in a single reservoir or 'lake'.

## 1 Introduction

Glacier lake outburst floods or jökullhlaups are a glacial hazard in many parts of the world (e.g. Björnsson, 1988; Clague et al., 2012). In addition, they provide a window into subglacial drainage systems: most outburst floods involve the opening and closing of a subglacial conduit, driven by melting of its walls through heat dissipation in the turbulent flow of water, and by viscous creep closure of the ice. In the early stages of the outburst flood, wall melting dominates through a positive feedback in which conduit enlargement leads to faster flow and therefore more dissipation of heat. Later in the flow, creep closure accelerates as the lake level drops, eventually causing the conduit to close again and terminating the flood.

While the mechanics of the main discharge phase of an outburst flood are relatively well-understood (Nye, 1976; Clarke, 1982, 2003; Fowler and Ng, 1996; Ng, 1998; Fowler, 1999; Kingslake and Ng, 2013; Kingslake, 2013), the mechanisms that initiate that flood are still poorly understood, even though they dictate the level to which the lake is able to fill, and therefore the magnitude of the flood. In other words, the challenge is to explain not how a single flood progresses once started, but how it starts, and more to the point, what makes repeated floods occur cyclically, as a self-sustaining oscillation in the drainage system (Fowler, 1999; Kingslake, 2015). The main discharge phase of the flood is generally only part of a larger cycle, which also comprise a recharge phase in which outflow from the lake is small or absent altogether, and crucially, the transition between the two, where the flood is initiated.



One possibility for flood initiation is that the lake simply fills to the level at which the ice starts to float on lake waters, and a sheet flow emerges between ice and bed that subsequently channelizes (Flowers et al., 2004). Some lakes are known to initiate their outburst floods before they reach that flotation level, which raises the question of what then starts the flow.

Motivated by the subglacial lake at Grímsvötn in Iceland, Ng (1998) and Fowler (1999) consider how outburst floods are initiated in a drainage system that consists of a Röthlisberger (R-) channel (Röthlisberger, 1972) controls the drainage and refilling of a lake over multiple Flood cycles. In the simplest version of their model, where the channel is fed purely by the lake, they find that the amplitude of floods grows from cycle to cycle, and negative pressures are eventually reached, meaning that floods should start through the flotation of the ice dam after a few cycles. At the heart of this behaviour is the fact that the R-channel can shrink to progressively smaller sizes between successive floods, making it harder to re-initiate drainage as the lake fills.

Grímsvötn typically starts its flood when water levels are below flotation (Björnsson, 1988). To explain this, Fowler (1999) considers the effect of a water supply along the length of the channel on its evolution. Such a water supply can maintain a minimum channel size even between outburst floods, with flow of water being directly directed partly into the lake and partly down-glacier to the margin, provided the glacier has a geometrical 'seal'. As the lake level fills, that intra-channel flow divide migrates towards the lake, and flood now initiates when the divide reaches the edge of the lake.

While this mechanism successfully explains how limit cycles (stable, periodic oscillations in lake level) can emerge in the model, it also predicts that no water can leave the lake between floods. Tracer experiments conducted elsewhere (Fischer, 1973) demonstrate that lakes can leak continuously throughout their recharge phase. In this paper, we consider an alternative mechanism by which floods can initiate in a recurring fashion without the need for a sealed lake, but with continuous leakage throughout the flood cycle. We show that a conduit that is able switch spontaneously between the behaviour of typical of an R-channel and the behaviour of a linked cavity system (Kessler and Anderson, 2004; Schoof, 2010; Hewitt et al., 2012; Hewitt, 2013; Werder et al., 2013) can sustain limit cycles without leading to flotation of the glacier at flood initation, and without requiring the flood divide migration appealed to in (Fowler, 1999).

This behaviour was first documented in a model by Schoof et al. (2014). In this paper, we approach the problem from a more theoretical perspective, focusing on the following: First, we delineate when a 'lake' or other storage reservoir can be drained steadily, and when steady drainage becomes unstable and when that instability leads to a limit cycle. Next, we investigate how the amplitude and period of floods depends on reservoir size and inflow rate. As a corollary, we determine at which point (in parameter space) the model predicts that flotation does occur and a different flood initiation mechanism is likely to take over. Lastly, motivated by an extension of classical jökullhlaups models to spatially spread-out water storage in Schoof et al. (2014), we investigate the mechanism by which behaviour resembling outburst floods can occur when storage is not localized, showing that the classical runaway melt effect first described by Nye (1976) can be replaced by another mechanism that relies on the introduction of a spatial phase shift between conduit size and water pressure along the flow path.

The original motivation for the work in Schoof et al. (2014) was to demonstrate that behaviour akin to outburst floods can occur in systems with limited or even distributed water storage, manifesting itself in the form of unforced water pressure oscillations. Viewed from that alternative different perspective, the present paper analyzes a drainage model that has become





widely adopted in the study of subglacial hydrology (Werder et al., 2013). We identifying instabilities that spontaneously lead to self-sustained oscillations, describing the mechanisms behind them and delineating the regions in parameter space where they occur. This is likely to be useful in diagnosing the behaviour of such models, regardless of their specific application to large outburst floods emanating from easily identifiable glacier 'lakes'.

To do so, we use a hierarchy of models, employing both, a spatially-extended, one dimensional drainage system model and a lumped, box-type model that provides additional physical insight, as well as being the appropriate limit of the spatially-extended model in the case of a long flow path. The paper is laid out as follows: in section 2, we develop the spatially extended and lumped models. In sections 3.1 and 3.2, we identify instabilities in the lumped model, and where they are suppressed. 'Instability' here refers to the system evolving away from a steady state if the latter is slightly perturbed. What that instability

evolves into is investigated in detail in sections 3.3 and 3.4, where we show that stable limit cycles emerge in the lumped model. In section 4, we show that the lumped model replicates the behaviour of the spatially extended behaviour well in the classical parameter regime that corresponds to the drainage of large glacier-dammed lakes. We also show that the extended model is unstable in more exotic parameter regimes, where the lumped model is not, and investigate the resulting dynamics. These results are summarized in section 5.1, and we take the results for localized reservoirs as motivation to study stability of

drainage systems with distributed storage in 5.2. Our results are summarized in section 6, and extensive additional information are provided in the supplementary material, including the code used in computations.

## 2   Model

### 2.1   A continuum model for outburst floods

We use the one-dimensional continuum model for drainage through a single conduit in Schoof et al. (2014), building on

similar models used elsewhere (e.g. Ng, 1998; Hewitt and Fowler, 2008; Schuler and Fischer, 2009; Schoof, 2010; Hewitt et al., 2012). Denoting conduit cross-section by $S(x,t)$, effective pressure by $N(x,t)$ and discharge by $q(x,t)$, where $x$ is downstream distance and $t$ is time, we put

$$\frac{\partial S}{\partial t} = c_1 q \Psi + v_o(S) - v_c(S,N), \tag{1a}$$

$$\frac{\partial S}{\partial t} + \frac{\partial q}{\partial x} = r c_1 q \Psi, \tag{1b}$$

$$q = q(S, \Psi), \tag{1c}$$

$$\Psi = \Psi_0 + \frac{\partial N}{\partial x}, \tag{1d}$$

on $0 < x < L$, subject to boundary conditions

$$-V_p(N)\frac{\partial N}{\partial t} = q_{in} - q \qquad\qquad\qquad \text{at } x = 0, \tag{1e}$$

$$N = 0 \qquad\qquad\qquad \text{at } x = L, \tag{1f}$$





where $L$ is the length of the flow path, and we assume the closures

$$v_o(S) = u_b h_r(1 - S/S_0), \qquad v_c(S, N) = c_2 S|N|^{n-1}N,$$

$$q(S, \Psi) = c_3 S^\alpha |\Psi|^{-1/2}\Psi. \tag{1g}$$

Here $c_1$, $c_2$, $c_3$, $n$, $\alpha$, $u_b$, $h_r$ and $S_0$ are positive constants with $\alpha > 1$, while $q_{in}$ is also prescribed as a function of time and $V_p > 0$ is a function of $N$. $\Psi_0$ is a geometrically determined background gradient, given in terms of ice surface elevation $s(x)$ and bed elevation $b(x)$ through

$$\Psi_0 = -\rho_i g \frac{\partial s}{\partial x} - (\rho_w - \rho_i)g \frac{\partial b}{\partial x}. \tag{1h}$$

Here $\rho_w$ and $\rho_i$ are the densities of ice and water, respectively, and $g$ is acceleration due to gravity. Note that effective pressure $N$ is linked to water pressures $p_w$, which is the primary observable in the field, through

$$N = p_i - p_w \tag{1i}$$

where $p_i$ is overburden (or more precisely, normal stress in the ice at the bed, averaged over a length scale much larger than that occupied by the channel). Typically (including in (1h)), $p_i$ is assumed to be cryostatic, equal to $\rho_i g(s - b)$.

Physically, the model represents a conduit whose size evolves due to a combination of dissipation-driven wall melting at a rate $c_1 q\Psi$, opening due to ice sliding over bed roughness at rate $u_b h_r(1 - S/S_0)$, and creep closure at rate $c_2 SN^n$, with discharge in the conduit given by a Darcy-Weisbach or Manning friction law as $c_3 S^\alpha |\Psi|^{-1/2}\Psi$ through (1g). $u_b$ is sliding velocity, $h_r$ is bed roughness and $S_0$ is a cut-off cavity size at which bed roughness is drowned out (Schoof et al., 2012). $S_0$ is typically a regularizing parameter that prevents conduits from becoming excessively large in regions where effective pressures are low, such as near the glacier terminus, but has little effect on conduit sizes along most of the flow path; this corresponds to the limit of large $S_0$.

Vanishing effective pressure at the end of the flow path $x = L$ simply reflects the assumption that overburden and water pressure vanish simultaneously (where we have neglected atmospheric pressure as a gauge throughout). The upstream boundary condition at $x = 0$ instead reflects water storage in a reservoir at the head of the conduit: (1e) represents conservation of mass in the reservoir, with inflow rate $q_{in}$, outflow at rate $q(0, t)$ through the conduit modelled by (1a)–(1d) and with water storage $V$ in the reservoir a function of effective pressure at $x = 0$ such that $dV/dN = -V_p$. Note that we deliberately use 'reservoir' rather than 'lake' here since we will be interested not only in large lake, but also in more modest-sized reservoirs as in Schoof et al. (2014). We will treat $V_p$ and $q_{in}$ as constants in most of what follows. Conservation of mass along the conduit (1b) by contrast assumes that water storage is negligible along the flow path. We show that the contribution of wall melting to mass flux is in general negligible in terrestrial drainage systems in section 2.2 of the supplementary material.

More generally, (1e) is justified as follows (Clarke, 2003). Water volume in the reservoir can be related to water depth $h_w$ and hence to water pressure $p_{w0}$ at the bottom of the reservoir as $dV/dh_w = A$, where $A$ is the surface area that corresponds



to the filling level $h_w$. With $p_{w0} = \rho_w g h_w$, $\mathrm{d}V/\mathrm{d}p_{w0} = A/(\rho_w g)$, In general, $A(h)$ is given by the shape of the reservoir. If we put $N = p_i - p_{w0}$ at the head of the conduit, then $h_w$ and therefore $A$ are functions of $N$ at $x = 0$. Equation (1e) follows with $V_p = A(h_w)/(\rho_w g)$; constant $V_p$ therefore corresponds to a lake with vertical sides, whose surface area is independent of filling level.

There is an important simplification we have made, irrespective of the particular choice of $V_p$: Our model effectively imposes an ice cliff at the conduit inlet $x = 0$, with the cliff typically above the flotation thickness. This ensures that $N(0,t)$ can change over time without the upstream end of the conduit migrating. This contrasts with a glacier that partially floats on the lake, in which case the upstream end of the conduit is always at $N = 0$, but migrates as the lake fills or empties, and the location of that upstream end rather than effective pressure is related to lake volume. The filling level in the lake no longer dictates the

effective pressure at the conduit inlet, but how much of the glacier has floated and therefore where that inlet has migrated to (see section 2.4 of the supplementary material).

The assumption of an ice cliff, in addition, allows us to assume that $\Psi_0 > 0$ along the entire flow path, without a 'seal' at a finite distance from the reservoir at which $\Psi_0$ changes sign (Fowler, 1999). With a seal, there is a finite region of negative $\Psi_0$ between reservoir and seal location, with water flow out of the reservoir possible only when local effective pressure gradients

$\partial N/\partial x$ are large enough to overcome the effects of the seal to make the total hydraulic gradient $\Psi$ positive in (1d). In the absence of a seal and without an ice cliff (so $N = 0$ at the edge of the reservoir), normal stress at the glacier bed just outside of the reservoir would then be below the flotation pressure, and there would be nothing to dam the reservoir (see section 2.4 of the supplementary material).

The two situations (an ice cliff and a seal) can be reconciled in the sense that a seal very close to the edge of the reservoir

corresponds to a short, steep ice surface slope rather than an actual cliff. This results in a large, negative $\Psi_0$ between reservoir and seal over a short distance, which can be balanced by an equally large $\partial N/\partial x$ over the same short distance, leading to a non-zero effective pressure at the seal itself, only a short distance from the lake.

Nevertheless, our simplifying assumption is relevant as the mechanism for initiating periodically recurring floods is fundamentally different from that in Fowler (1999): his model relies on a geometrical seal with negative $\Psi_0$ near the reservoir,

changing sign at the seal location, and an englacial water supply to the conduit that dictates the gradient $\partial N/\partial x$ while the lake is filling. As described in the introduction, the onset of the outburst flood corresponds to the instant at which $\Psi$ at the the conduit inlet $x = 0$ changes from negative to positive, and allows water to flow out of the reservoir. This leads to runaway enlargement of the conduit through dissipation-driven melting.

By contrast, our model assumes that the reservoir always experiences some amount of leakage, and requires no englacially-

supplied conduit: the water supply to the drainage system in our model is purely through the inflow term $q_{in}$ into the reservoir. Leakage out of the reservoir is facilitated by the linked-cavity behaviour of the conduit between floods, associated with the cavity opening term $v_o$ that keeps the conduit open even when flow rates are small. This is absent from the models in Fowler (1999) and Ng (1998). As the lake fills, these cavities enlarge because the effective pressure is reduced, and the conduit then grows through the same melt-driven channel enlargement as in Fowler (1999) and Ng (1998), but the initiation of that

enlargement differs.





## 2.2 A lumped model

The model (1) can be simplified if we assume that $\Psi_0$ not only does not change sign along the flow path, but can be treated as a constant, and if we assume that the effective pressure gradient $\partial N/\partial x$ along the flow path can be treated as negligible (see also Fowler, 1999; Ng, 2000, and sections 2.2 and 2.3 of the supplementary material).

Under these assumptions, we can relate the flux $q$ along the flow path (which is independent of position by (1b)) purely to the conduit size $S$ and hydraulic gradient $\Psi_0$ at the head of the channel, leading to a system of ordinary rather than partial differential equations:

$$\dot{S} = c_1 q \Psi + v_o(S) - v_c(S, N) \tag{2a}$$

$$-V_p \dot{N} = q_{in} - q \tag{2b}$$

$$q = q(S, \Psi) \tag{2c}$$

where a dot signifies an ordinary derivative with respect to time, and we continue to assume the closure relations (1g); by the argument above, we then strictly speaking have to put $\Psi = \Psi_0$. We generalize this reduced model slightly to account qualitatively, if not quantitatively, for the effect of pressure gradients along the flow path by putting

$$\Psi = \Psi_0 - N/L, \tag{2d}$$

which corresponds to a crude divided difference approximation of the actual gradient $\partial N/\partial x$ by $[N(L,t) - N(0,t)]/L$

In what follows, we proceed first with an analysis of the simpler, 'lumped' model (2), to which we can bring to bear the theory of finite-dimensional dynamical systems (Wiggins, 2003), leading a number of semi-analytical results. Subsequently, we study the full, spatially-extended model (1), for which we are limited to comparatively expensive numerical methods that we can guide by our analysis of the simpler, lumped model.

## 3  An analysis of the lumped model

### 3.1  Nye's jökulhlaup instability in the reduced model

Nye's (1976) original theory of jökulhlaups centers on the idea that steady flow in a channel is unstable if there is a water reservoir that keeps water pressure approximately constant. This instability results in the runway growth of a channel, eventually allowing the reservoir to drain in an outburst flood. The assumption of a constant effective pressure is of course a simplification

of the more complete model (2), based on the notion that the drainage of the lake is generally too slow to lead to changes in effective pressure that could stabilize the channel against growth.

The basis of this instability is relatively easy to capture mathematically. To set the scene, we present a simplified version first before tackling an analysis of the complete mdel (2). If we assume a classical 'channel' in the sense of Röthlisberger (1972) and Nye (1976) and therefore set the cavity opening term $v_o$ to zero in (2a), and use the remaining relations in (1g), the conduit





evolution equation (2a) becomes

$$\dot{S} = c_1 c_3 S^\alpha |\Psi|^{3/2} - c_2 S |N|^{n-1} N, \tag{3}$$

with a steady state conduit size $\bar{S}$ given by $c_2 \bar{S} |\bar{N}|^{n-1} \bar{N} = c_1 c_3 \bar{S}^\alpha |\Psi|^{3/2}$. At fixed effective pressure $N$, and hence at fixed $\Psi$, this steady state is unstable: an increase $S'$ away from the steady state size $\bar{S}$ will lead to a further growth in he conduit, as

we have $(\bar{S} + S')^\alpha \approx \bar{S}^\alpha + \alpha \bar{S}^{\alpha-1} S'$ and so

$$\dot{S}' \approx c_1 c_3 \alpha \bar{S}^{\alpha-1} S' |\Psi|^{3/2} - c_2 S' |N|^{n-1} N$$
$$= c_1 c_3 (\alpha - 1) \bar{S}^{\alpha-1} |\Psi|^{3/2} S', \tag{4}$$

where, with $\alpha > 1$, the right-hand side is positive if $S'$ is, signifying that $S'$ will continue to grow in a positive feedback.

   In more abstract terms, if $\Psi$ and $N$ are fixed, (2a) can be expected to lead to unstable growth of conduits away from an

equilibrium state $\bar{S}$ if (see also pages 5–6 of the supplementary material to Schoof (2010))

$$c_1 \left. \frac{\partial q}{\partial S} \right|_{S=\bar{S}} \Psi + \left. \frac{\partial v_o}{\partial S} \right|_{S=\bar{S}} - \left. \frac{\partial v_c}{\partial S} \right|_{S=\bar{S}} > 0. \tag{5}$$

This is also precisely the condition that defines a conduit as being 'channel-like' in Schoof (2010), in the sense that two neighbouring conduits will compete for water with one eventually growing at the expense of the other if both are 'channel-like'. A channel-like conduit is also distinguishable by the fact that the effective pressure required to balance opening processes (dissipation-driven wall melting and cavity opening due to flow over bed toughness) by creep closure increases with discharge

$q_{in}$ in the channel; the opposite is true for a 'cavity-like' conduit.

   Key to the instability mechanism above was the notion that effective pressure $N$ is kept constant as the conduit evolves, which is not actually the case: instead, the lake simply buffers changes in $N$ from happening rapidly, since a change in $N$ can only occur after an imbalance develops between inflow $q_{in}$ and outflow $q$. Next, we investigate in more detail how water

storage and the finite size $L$ of the system control whether Nye's instability does occur in the reduced model of section 2.2. Our main result will be that this is not unconditionally the case, and that some reservoirs can drain steadily without outburst floods.

   Note that $v_o$ and $v_c$ satisfy

$$\frac{\partial v_o}{\partial S} \le 0, \qquad \frac{\partial v_c}{\partial S} > 0, \qquad \frac{\partial v_c}{\partial N} > 0 \tag{6}$$

with $v_o(S) > 0$ bounded as $S \to 0$, $v_c(S,0) = 0$ and $v_c(0,N) = 0$, while $q$ satisfies

$$\frac{\partial q}{\partial S} > 0, \qquad \frac{\partial q}{\partial \Psi} > 0. \tag{7}$$

$q$ also has the same sign as $\Psi$, and satisfies $q(0,\Psi) = q(S,0) = 0$. These are the minimal assumptions we make on these functions, allowing us to generalize from the specific forms in (1g) in analyzing Nye's instability.





The primary dependent variables in the model (2) are $S$ and $N$. For constant water input $q_{in}$, the model admits a steady state $(\bar{S}, \bar{N})$ given implicitly by

$$c_1 q(\bar{S}, \bar{N})\bar{\Psi} + v_o(\bar{S}) - v_c(\bar{S}, \bar{N}) = 0 \tag{8a}$$

$$q(\bar{S}, \bar{\Psi}) = q_{in} \tag{8b}$$

$$\bar{\Psi} = \Psi_0 - \bar{N}/L. \tag{8c}$$

If $q_{in} > 0$, it follows that $\bar{\Psi} > 0$, and from (8a) and the properties of $v_c$, we also have $\bar{N} > 0$. For future convenience, we also write $q(\bar{S}.\bar{N}) = \bar{q}$. Note that, for the specific choices in (1g), a steady state exists for every positive $q_{in}$. Eliminating $\bar{S}$ and $\bar{\Psi}$, we find a problem for $\bar{N}$ alone:

$$c_1 q_{in}\left(\Psi_0 - \frac{\bar{N}}{L}\right) + v_0 - c_2\left(\frac{q_{in}}{c_3(\Psi_0 - \bar{N}/L)}\right)^{1/\alpha} \bar{N}^n = 0$$

The left-hand side is a monotonically decreasing function of $\bar{N}$ for $0 \leq \bar{N} < \Psi_0 L$, positive at $\bar{N} = 0$ and tending to $-\infty$ as $\bar{N} \to \Psi_0 L$, so the equation always admits a unique solution for $\bar{N}$ in this range, from which $\bar{S}$ can then be determined. The solution then has positive $\bar{N}$, $\bar{S}$ and $\bar{\Psi}$.

The point of the stability analysis is ultimately to establish conditions under whether this steady state is stable and can therefore persist over time: if so, no outburst floods need to result from the presence of the reservoir. Linearizing about the steady state as

$$N = \bar{N} + N', \qquad S = \bar{S} + S'$$

and putting $\bar{V}_p = V_p(\bar{N})$ gives the following leading-order form of (2):

$$\dot{S}' = c_1 q_S \bar{\Psi} S' - (q_\Psi \bar{\Psi} + \bar{q}) L^{-1} N'$$

$$+ v_{o,S} S' - v_{c,S} S' - v_{c,N} N' \tag{9a}$$

$$-\bar{V}_p \dot{N}' = -q_S S' + q_\Psi L^{-1} N' \tag{9b}$$

where

$$q_S = \left.\frac{\partial q}{\partial S}\right|_{S=\bar{S}, \Psi=\bar{\Psi}}, \qquad q_\Psi = \left.\frac{\partial q}{\partial \Psi}\right|_{S=\bar{S}, \Psi=\bar{\Psi}}, \qquad v_{o,S} = \left.\frac{\mathrm{d}v_o}{\mathrm{d}S}\right|_{S=\bar{S}},$$

$$v_{c,S} = \left.\frac{\partial v_c}{\partial S}\right|_{S=\bar{S}, N=\bar{N}}, \qquad v_{c,N} = \left.\frac{\partial v_c}{\partial N}\right|_{S=\bar{S}, N=\bar{N}}.$$

As before, we want to know whether $S'$ grows over time. Looking for solutions of the form $S' = S'_0 \exp(\lambda t)$, $N' = N'_0 \exp(\lambda t)$, we get the eigenvalue problem

$$\begin{pmatrix} c_1 q_S \bar{\Psi} + v_{o,S} - v_{c,S} - \lambda & -c_1(q_\Psi \bar{\Psi} + \bar{q})L^{-1} - v_{c,N} \\ \bar{V}_p^{-1} q_S & -\bar{V}_p^{-1} q_\Psi L^{-1} - \lambda \end{pmatrix} \begin{pmatrix} S'_0 \\ N'_0 \end{pmatrix} = 0$$

Setting the determinant of the matrix on the left to zero leads to a polynomial for $\lambda$,

$$\lambda^2 - a_1 \lambda + a_2 = 0 \tag{10}$$





where the coefficients take the form

$$a_1 = c_1 q_S \bar{\Psi} + v_{o,S} - v_{c,S} - \bar{V}_p^{-1} q_\Psi L^{-1} \tag{11a}$$

$$a_2 = \bar{V}_p^{-1} q_S \left[ c_1 (q_\Psi \bar{\Psi} + \bar{q}) L^{-1} + v_{c,N} \right]$$
$$\qquad - \bar{V}_p^{-1} q_\Psi L^{-1} \left[ c_1 q_S \bar{\Psi} + v_{o,S} - v_{c,S} \right]$$

$$\qquad = \bar{V}_p^{-1} \left[ c_1 q_S \bar{q} L^{-1} + q_S v_{c,N} + q_\Psi L^{-1} (v_{c,S} - v_{o,S}) \right] \tag{11b}$$

But, from our assumptions on the various functions involved, we see that $a_2 > 0$ (recall that $v_{o,S} \leq 0$ from (6)), while $a_1$ can be either sign.

The characteristic quadratic (10) has solutions

$$\lambda = \frac{1}{2} \left[ a_1 \pm \sqrt{a_1^2 - 4a_2} \right] \tag{12}$$

Since $a_2 > 0$, we have $a_1^2 - 4a_2 < a_1^2$ and two possible types of solution: either $a_1^2 - 4a_2 > 0$ and we have two real roots, both of which have the same sign as $a_1$. Alternatively, we have $a_1^2 - 4a_2 < 0$ and a complex conjugate pair of roots, both of which have real part $a_1$. In either case, we see that the system is linearly unstable if and only if $a_1 > 0$, or

$$\left( c_1 q_S \bar{\Psi} + v_{o,S} - v_{c,S} \right) - \bar{V}_p^{-1} q_\Psi L^{-1} > 0. \tag{13}$$

We have deliberately written the left-hand side of (13) as the difference of two terms, a potentially destabilizing term
$c_1 q_S \bar{\Psi} + v_{o,S} - v_{c,S}$ and a stabilizing term $-\bar{V}_p^{-1} q_\Psi L^{-1}$. The first term can be recognized as the growth rate sensitivity we previously identified as being at the heart of Nye's instability in (5), as well as an indicator of whether the steady-state conduit is 'channel-like' in the terminology of Schoof (2010).

The second term in (13) is a stabilizing term that is inversely related to storage capacity. Its physical origin is the following: the sensitivity of channel growth to perturbations in conduit size may be positive, potentially leading to unstable conduit growth.
However, growth of the conduit also allows water to drain out of the system, which will increase the effective pressure $N$. As the effective pressure is increased, the hydraulic gradient $\Psi = \Psi_0 - N/L$ is reduced. This leads to less turbulent dissipation in the conduit as the conduit grows, and increased $N$ further leads to faster creep closure of the channel. Both of these will suppress further growth of the conduit.

How strong this stabilizing effect is will depend on the storage capacity of the system. If the storage capacity is large, then
the growth of the channel will have a minimal impact on effective pressure (essentially, the amount of water that drains out due to the widened conduit is insufficient to affect water levels and therefore water pressure in the storage system sufficiently). The stabilizing effect of the second term is therefore small when storage capacity $\bar{V}_p$ is large. Similarly, if the system size $L$ is large, then the role of effective pressure in controlling hydraulic gradients is small, and $\Psi$ is always close to the background hydraulic gradient. The stabilizing effect is then again small, as drawing water out of the storage system does not significantly
affect hydraulic gradients and turbulent dissipation.

In summary, Nye's instability will occur if the conduit is channel-like and water storage in the system is sufficiently large, and if the system length $L$ is big enough. Note that with the choices in (1g), the system is always unstable if $v_o = 0$ (so the





| Parameter | value |
|---|---|
| $c_1$ | $1.3455 \times 10^{-9}$ J$^{-1}$ m$^3$ |
| $c_2$ | $3.44 \times 10^{-24}$ Pa$^{-3}$ s$^{-1}$ |
| $c_3$ | $4.05 \times 10^{-2}$ m$^{9/4}$ Pa$^{-1/2}$ s$^{-1}$ |
| $\alpha$ | 5/4 |
| $n$ | 3 |
| $u_b h_r$ | $3.12 \times 10^{-8}$ m$^2$ s$^{-1}$ |
| $S_0$ | 170 m$^2$ (spatially extended model) |
| $S_0$ | $\infty$ (lumped model) |
| $\Psi_0$ | 178 Pa m$^{-1}$ |

**Table 1.** Parameter values common to all calculations except those in figure 11, for which $\Psi_0 = 1630$ Pa m$^{-1}$ (corresponding to a much steeper 17:100 slope), $u_b h_r = 1.05 \times 10^{-07}$ m$^2$ s$^{-1}$ and $L = 5$ km.

conduit is always a channel) and $L = \infty$ (so effective pressure does not alter the hydraulic gradient). This is in agreement with Ng (1998).

## 3.2 Stability boundaries

So far, we have only established abstract conditions under which steady drainage through the conduit is unstable and will

therefore not persist, potentially leading to periodic outburst floods. We can go further and determine explicitly the regions of parameter space in which this stability occurs. While we present our results here in dimensional form, note that it is possible to reduce the spatially extended and lumped models (1) and (2) to a four-dimensional parameter space by non-dimensionalizing them (sections 2.2 and 2.3 of the supplementary material). These parameters are dimensionless versions of the inflow rate $q_{in}$, storage capacity $V_p$, system length $L$ and conduit cut-off size $S_0$. Recall that $S_0$ is intended to have minimal impact on

conduit evolution away from the glacier margin, and we are therefore interested in the limit of large $S_0$. As a result, we restrict ourselves to the three-dimensional parameter space spanned by $(q_{in}, \bar{V}_p, L)$ and set $S_0 = \infty$ in the lumped model (2). The remaining fixed parameter values we have used are given in table 1. The low value of $\Psi_0$ stated corresponds approximately to a 1:50 surface slope.

It can be shown analytically that there is at most a finite range of values of $q_{in}$ for which the instability occurs for given $\bar{V}_p$

and $L$ (section 3.1 of the supplementary material): when $q_{in}$ is too small, the conduit is cavity-like as opposed to channel-like, so (5) is not satisfied. By contrast, when $q_{in}$ is too large, the stabilizing term $\bar{V}_p^{-1} q_\psi L^{-1}$ in (13) dominates. The intermediate range of inflow values $q_{in}$ that makes the system unstable is not guaranteed to exist: a sufficiently large reservoir size $\bar{V}_p$ and a long flow path length $L$ are required to prevent the stabilizing term from dominating. The range of unstable $q_{in}$ values also increases as $\bar{V}_p$ and $L$ increase.

This is confirmed by direct numerical computations of the stability boundaries. These are the locations in parameter space where the real part of the growth rate $\lambda$ in (12) is zero (see section 3.2 of the supplementary material for details): figure 1 shows

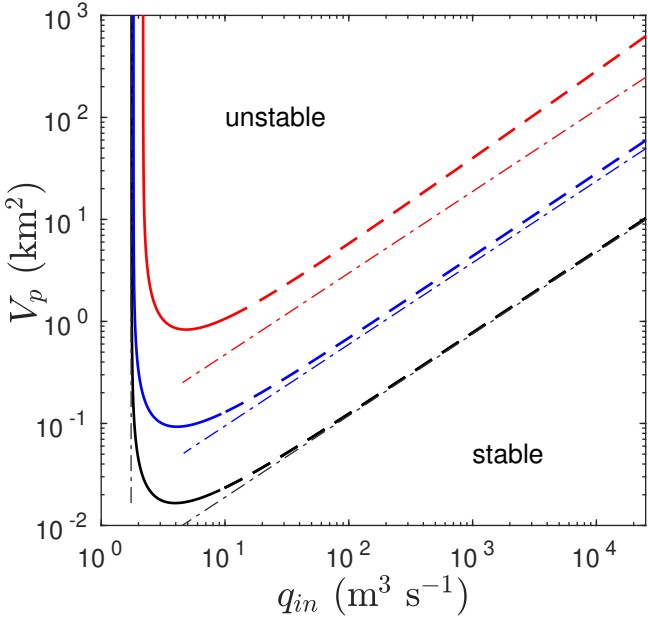

**Figure 1.** Stability boundaries for lake systems with the parameter values in table 1 shown as solid/dashed curves. Note the logarithmic scales on the axes. The unstable region in parameter space is above the curves as shown. The curves correspond to different values of $L$: 10 km (red), 50 km (blue) and 250 km (black); the latter is not intended to be physically realistic but to reflect the limit of large $L$. The Hopf bifurcation at the stability boundary is supercritical where the curve is solid, subcritical where dashed (see section 3.3. The dot-dashed curves show the asymptotic stability boundaries (14) and (15).

stability boundaries in the $(q_{in}, \bar{V}_p)$-plane for different values of $L$. Note that $\bar{V}_p$ is displayed in units of km$^2$, normalizing by $\rho_w g$; in other words, what is really plotted is $\bar{V}_p/(\rho_w g)$, the surface area of the lake (see section 2). Similarly, we will plot $N$ in units of metres, normalizing by $\rho_w g$: changes in $N$ plotted then correspond directly to changes in water level in the lake.

In each case, there is a region of instability above some critical value of $\bar{V}_p$, and for $q_{in}$ in some intermediate range, where

5 larger $\bar{V}_p$ is required and the unstable inflow range is shrunk for shorter flow path lengths $L$.

For large enough $\bar{V}_p$ and $L$, we can determine analytical expressions for the stability boundaries. The lower critical value of $q_{in}$ at which the drainage system first becomes unstable corresponds to the switch from a cavity-like to a channel-like conduit, and this occurs for large $L$ at (see Schoof, 2010, and section 3.1 of the supplementary material)

$$q_{in} = \frac{u_b h_r}{c_1(\alpha - 1)\Psi_0}. \tag{14}$$

10 The upper critical value of $q_{in}$ at which the system stabilizes again can similarly computed in the limit of large $L$ by omitting the cavity opening term $v_0$ and reducing conduit evolution (2a) to a balance between the first (dissipation-driven melting) and





third (creep-closure) terms on the right hand side. This yields

$$\bar{V}_p \sim \frac{q_{in}^{1/\alpha}}{2(\alpha-1)c_1 c_3^{1/\alpha}\Psi_0^{(3\alpha+2)/(2\alpha)}L}. \tag{15}$$

These limiting forms are also shown in figure 1, where we see that they are most accurate for large $L$ as expected.

### 3.3 Hopf bifurcations and limit cycles

The analysis above has been purely linear, identifying parameter regimes in which steady drainage is unstable. What the analysis does not allow is to say is what happens when the perturbations grow in size to the point where the linearization fails.

A key aspect of many outburst floods is that they are a recurring phenomenon; in terms of our reduced model (2) with two dynamical degrees of freedom and steady forcing, such recurrence must correspond to a stable periodic oscillation in the absence of time-dependent forcing (see also Kingslake, 2013, for time-dependent forcing that leads to chaotic solutions). As

was underlined by Ng (1998) and Fowler (1999), the existence of such a limit cycle does not simply follow from the instability itself: we need to ensure that the evolution away from the steady state leads to bounded growth of the instability once it reaches a finite amplitude (as opposed to the infinitesimal perturbation assumed by the linearization in section 3.1). This cannot be done in the confines of the linearization of section 3.1 alone.

It is straightforward to demonstrate bounded growth in our model computationally. Figure 2 shows a sample calculation of a

periodic solution, with the classical attributes of a outburst flood cycle: effective pressure slowly decreases during the interval between outburst floods, when conduit size is small. This is simply the lake refilling. As $N$ approaches its minimum, the conduit size $S$ starts to grow rapidly, initiating the outburst flood: effective pressure is no longer large enough to keep conduit size small, and enough water can flow to start enlarging the conduit in the runaway growth envisioned by Nye (1976). The lake then drains, rapidly increasing $N$. Once effective pressure gets large enough, creep closure becomes dominant, causing $S$ to

shrink again, and the cycle repeats. By contrast with Ng (1998) and Fowler (1999), key to this periodic behaviour is that the conduit cannot become arbitrarily small between floods, since it is kept open by ice flow over bed roughness.

In fact, $S$ in our model actually starts to increase immediately after flood termination, as the refilling of the reservoir leads to decreasing effective pressure allowing the now cavity-like conduit to grow. As explained above, water flow through them will eventually lead to re-initiation of dissipation-driven melting and enlargement of the conduit. By contrast, once the amplitude of

the floods has become large enough, conduit size in the pure channel model of (Ng, 1998) and (Fowler, 1999) keeps shrinking during the refilling phase until the effective pressure changes sign to negative, and this underpins the ever-increasing flood amplitudes in their model.

An alternative and possibly preferable way to visualize the periodic solution is in a phase plane, plotting $S$ against $N$ as the system evolves; a periodic solution then corresponds to a closed orbit. Figure 3 shows the phase plane equivalent of (2), with

the dashed lines corresponding to nullclines (the curves on which either $\dot{S}=0$ or $\dot{N}=0$. During the refilling phase, the point $(N(t), S(t))$ closely follows the $S$-nullcline, with $S$ steadily increasing as explained above.

Figures 2 and 3 show only one example, and the question remains as to whether closed orbits invariably result from the instability of section 3.1, and how these orbits change as parameter values are changed. It is possible to solve directly for such

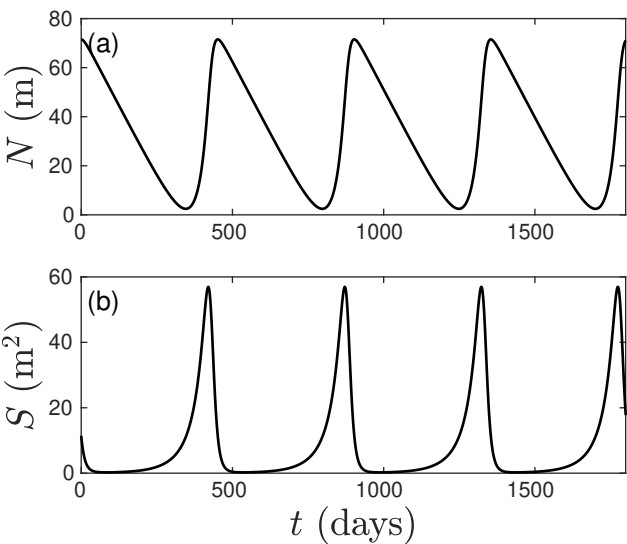

**Figure 2.** Periodic oscillations in the system (2) with parameter values as in table 1, with $V_p = 4$ km$^2$, $L = 50$ km and $q_{in} = 10.9$ m$^3$ s$^{-1}$.

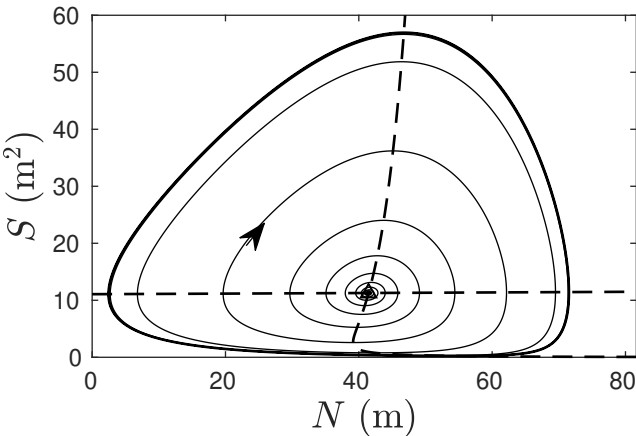

**Figure 3.** The periodic solution of figure 2 plotted in a phase plane. The thin curve represents a solution that approaches the limit cycle shown as a thick curve as indicated by the arrow, and the instability of the steady state solution (where the nullclines cross) is clearly visible, since evolution along the thin line is away from that steady state.





periodic solutions and trace how they change under parameter changes using an arc length continuation method (section 3.4 of the supplementary material), and to determine simultaneously whether the periodic solutions are stable: this is not guaranteed, and some periodic solutions are never attained by forward integration of (2) because small perturbations will cause the system to evolve away from them.

As in section 3.2, we focus on how solutions change as the water input to the reservoir $q_{in}$ is varied. Figure 4 shows how the amplitude of oscillations varies with water input for a fixed $L = 50$ km and different storage capacity $V_p$ (treated as independent of $N$ here). In each panel, the coloured curve (black in panel e) shows the minimum and maximum value of $N$ attained for a periodic solution at the corresponding value of $q_{in}$ (i.e., the values of $N$ where the corresponding orbit in the phase plane crosses the $N$-nullcline). Solid portions of these coloured curves in figure 4 correspond to stable periodic solutions, while

dotted portions are unstable periodic orbits. The black curves in figure 4 generally correspond to steady state solutions, plotting $N$ in the steady state against $q_{in}$ (we use black for steady states and oscillatory solutions in panel e, but these are easily distinguished by comparing the plots in panel e with those in the remaining panels). Again, solid black lines are stable steady states and dashed black lines are unstable steady states. Each panel in figure 4 corresponds to a different storage capacity $V_p$ and therefore represents a horizontal slice through panel figure 1; $V_p$ decreases from panel a to panel d.

As $q_{in}$ crosses the lower critical value at which instability first occurs (associated with the change in the steady state conduit from cavity- to channel-like behaviour), a limit cycle of small amplitude is formed, with that amplitude growing progressively as $q_{in}$ increases. This is consistent with a supercritical Hopf bifurcation: in general, the change from stability to instability in our model corresponds to the eigenvalue $\lambda$ in (12) attaining the purely imaginary value $\pm i\sqrt{a_2}$, and a standard result from the theory of dynamical systems is that a local, small-amplitude periodic solution exists near the bifurcation (Wiggins, 2003). What

is non-trivial to determine *a priori* is whether that periodic oscillation exists on the stable or unstable side of the bifurcation (in this, for values of $q_{in}$ less than or larger than the critical value, respectively). The stability of the periodic solution complements that of the corresponding steady state: if the periodic solution appears on the side of the bifurcation where the steady state has become unstable (a case termed a supercritical Hopf bifurcation), then the periodic solution is stable. This is the case for the lower critical value of $q_{in}$ in all panels of figure 4.

As the upper critical value of $q_{in}$ is approached, the stability analysis of section 3.1 predicts that the steady state returns to stability. As this happens, we see that the amplitude of oscillations only shrinks continuously back to zero for the smallest value of $\bar{V}_p$ considered (in which case the upper critical value corresponds to another supercritical Hopf bifurcation). In panels a-c, the amplitude of stable periodic solutions continues to increase until values of $q_{in}$ near the upper critical value are reached. The amplitude then decreases slightly with a further increases in $q_{in}$, before the periodic solution ceases to exist at a third critical

value of $q_{in}$ that is larger than the threshold at which the steady state has become stable again, as shown in more detal in the inset in panel b. (In technical terms, this is known as a saddle-node bifurcation of the Poincaré map of the dynamical system (2), see Wiggins (2003).)

    Where this abrupt disappearance of the stable periodic solution occurs, it is generally accompanied by the existence of an unstable periodic solution near the upper critical value of $q_{in}$ at which the steady state becomes stable again (see in particular

the inset in panel b). That critical value generally corresponds to a so-called subcritical Hopf bifurcation (as a counterpart to

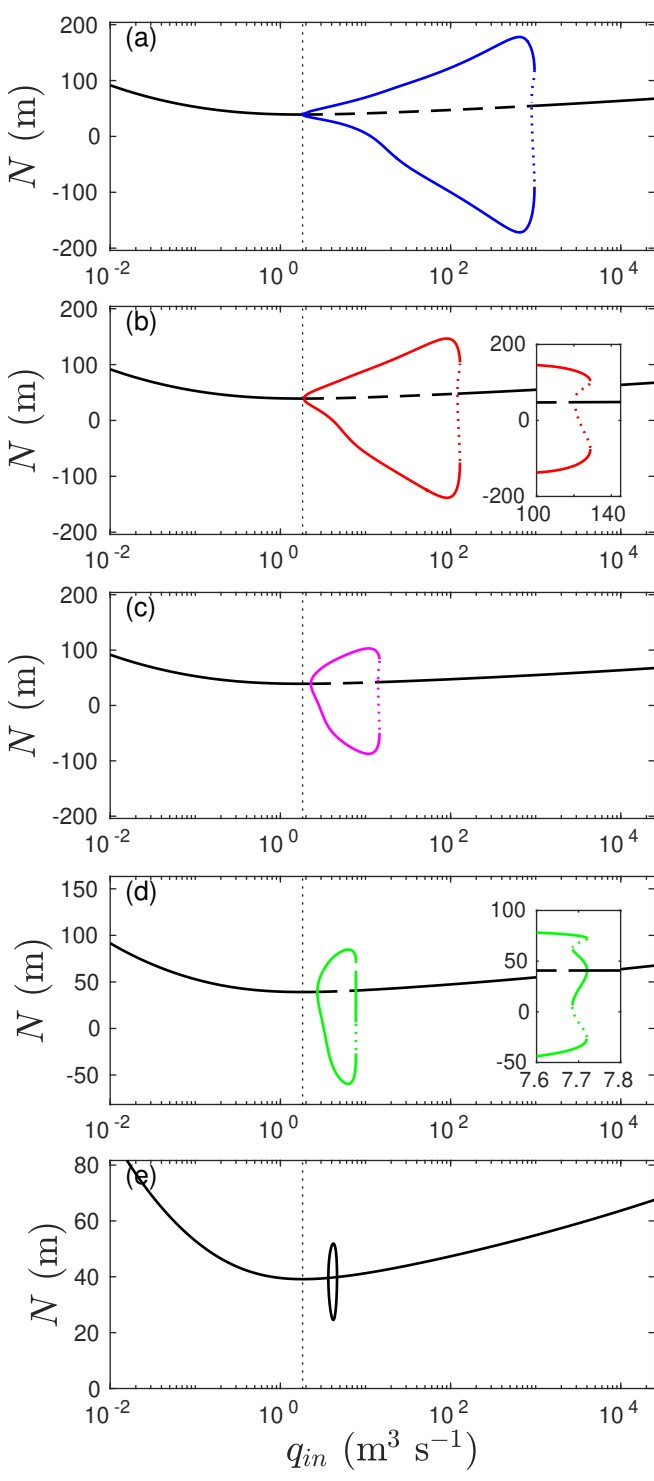

**Figure 4.** Bifurcation diagrams for $L = 50$ km, $V_p = 4$ (a), 0.8 (b), 0.16 (c), 0.112 (d), 0.094 (e) km$^2$. Plotted are stable (solid black) and unstable (dashed black) steady state $N$, minimum and maximum $N$ attained in stable (solid coloured) and unstable (solid dashed) periodic solutions. The insest in (b) and (d) show details, while the vertical dotted line marks the value of $q_{in}$ at which the conduit becomes channel-like.





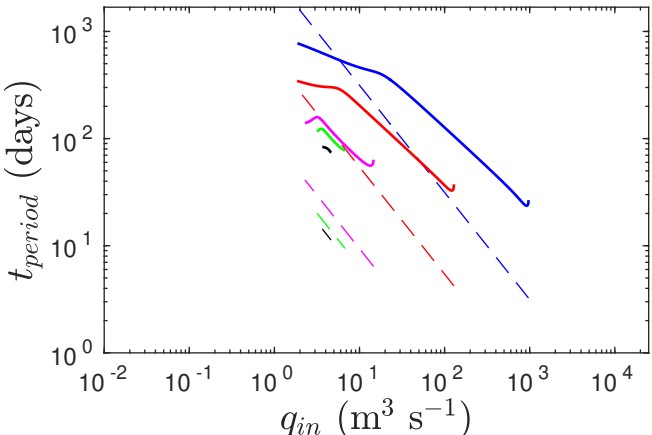

**Figure 5.** The period of stable periodic solutions where they exist for the same parameter values as in figure 4. The colour scheme is the same as for the coloured curves in figure 4, where only the stable portion of the periodic solution curves is shown. The dashed line here represent the asymptotic solution for the period of oscillation, (16)

its supercritical cousin that we encountered above). Panel d (see inset) represents an exotic exception to this, where there are two stable periodic solutions for a small interval of $q_{in}$.

The nature of the Hopf bifurcations can be determined more quickly than by the numerical continuation method used above, using a weakly nonlinear stability analysis in the vicinity of the stability boundary (see section 3.3 of the supplementary

material). This allows us to map out in figure 1 where that boundary corresponds to a supercritical Hopf bifurcation, in whose vicinity a small-amplitude stable periodic solution emerges (stability boundary indicated as a solid curve), and where the Hopf bifurcation is subcritical and the stable periodic solution has a large amplitude. As in the few samples shown in figure 4, we see that the lower critical value of $q_{in}$ is always supercritical, while the upper critical value is generally subcritical, except at low $V_p$.

In practical terms, figure 4 shows that for every unstable steady state, there is a corresponding stable periodic solution, so the system evolves away from the steady state into an oscillation of finite amplitude as shown in figures 2 and 3. In practice, we may wish to know not only how the amplitude of oscillations (that is, of variations in $N$ during the flood cycle) depends on water supply to the reservoir, but also what the recurrence period of the floods is. Figure 5 shows the period of the stable periodic solutions shown in figure 4 as a function of water supply rate $q_{in}$, using the same colouring scheme to distinguish

different values of $V_p$. Perhaps unsurprisingly, large inflow rates $q_{in}$ correspond to more rapid flood cycles, and large reservoir volumes correspond to floods repeating more slowly, albeit with a larger amplitude.

There is a significant caveat here: in many cases, the limit cycle solutions we have computed predict that $N$ becomes negative during the cycle of reservoir filling and draining. This is not something that the model (2), or indeed its spatially extended counterpart (1), were intended to capture. Instead of the creep closure of the conduit $v_c(S, N)$ simply becoming

negative and the conduit 'creep-opening', we expect that the glacier should instead detach from the bed and a sheet flow of

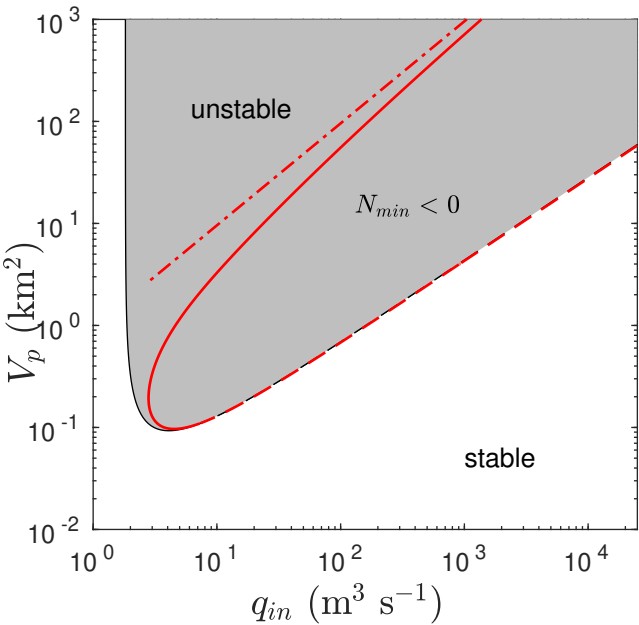

**Figure 6.** The stability boundary of figure 1 for $L = 50$ km plotted in black, with the region of parameter space in which the steady state is unstable shown in grey. The red curve is set of parameter combinations $(q_{in}, V_p)$) for which there is a periodic solution that reaches a minimum effective pressure of $N = 0$. The minimum effective pressure becomes negative inside the region delineated by the red curve, as well as in a small strip to the right of the right-hand branch of the red curve. The dot-dashed curve is the asymptotic formula (17) for the threshold value of $q_{in}$ at which negative $N$ is first attained.

water initiates the outburst flood in this case; prototypes of models for this eventuality can be found in Schoof et al. (2012), Hewitt et al. (2012) and Tsai and Rice (2010).

It is important to know where in parameter space the outburst flood mechanism will change away from conduit growth due to cavities becoming channel-like at the end of the refilling phase, and instead involve water separating ice from bed at vanishing effective pressure. The boundary between these two regimes should correspond to the location in parameter space where the minimum effective pressure during the flood cycle is zero. Figure 6 shows that parameter regime boundary, computed numerically (section 3.4 of the supplementary material) and superimposed on the stability boundary plot of figure 1. Clearly, sheet-flow-initiated floods (in which the glacier starts to float at the start of the outburst flood) are favoured at high water input rates and smaller reservoir volumes, where the conduit is less able to adjust to the rapid refilling of the reservoir.

## 3.4 Asymptotic solutions

We can also address limit cycle solutions through asymptotic methods in some parametric limits in our model. The most relevant limit for 'real' glacier-dammed lakes is likely to be that of a relatively large reservoir that is filled relatively slowly,



but where water supply is not so small as to allow the conduit to be cavity-like in steady state (in which case the reservoir would be drained steadily, without a flood cycle): This is the case of large $V_p$ and moderate but not small $q_{in}$, and is described in appendix A and, in detail, in section 4 of the supplementary material. This region of parameter space lies in the upper left of the unstable region of figure 6, between the near-vertical solid black line and the solid red curve.

In brief, the asymptotic solution confirms that there is a periodic flood cycle that the system very quickly settles into, and that the flood cycle consists of three distinct stages. During the main flood stage, the evolution of the conduit is rapid and dominated by dissipation-driven melt $c_1 q \Psi$ and creep closure $v_c(S, N)$ in (2a), and water input $q_{in}$ to the lake is much smaller than outflow $q$. This is followed by a long refilling phase in which the conduit has shrunk dramatically and therefore behaves as a cavity. Its size is dictated by a quasi-equilibrium in which the cavity opening term $v_o(S)$ balances the creep closure term

$v_c(S, N)$, leading to a slow opening of the conduit as the reservoir fills and $N$ consequently dropped. Outflow $q$ from the reservoir during this phase is insignificant, and the mass balance of the lake is dominated by inflow $q_{in}$. The refilling phase is terminated by a flood initiation phase whose length is intermediate between the main refilling phase and the rapid flood phase. During this initiation phase, the reservoir is still filling, but the conduit has enlarged sufficiently that dissipation-driven wall melting $c_1 q \Psi$ starts to be significant.

Qualitatively, this solution is illustrated by the limit cycle in figures 2 and 3 and panel a of figure 8. The asymptotic solution only becomes quantitatively accurate when $V_p$ is large enough to be physically unrealistic for real glacier-dammed lakes (section 4.4 of the supplementary material); this is because the relatively large exponent $n = 3$ in Glen's law makes the creep closure term quite sensitive to changes in $N$ when $N$ is small, and the initiation phase of the floods is affected significantly by this.

    One of the predictions of the asymptotic solution is that the amplitude of effective pressure oscillations should be insensitive to refilling rate $q_{in}$, except close to the Hopf bifurcation at which oscillations are initiated. As a result, the flood recurrence period (which is essentially the time taken to refill the reservoir in the limit where reservoir drainage is fast) should simply be inversely proportional to $q_{in}$. Specifically, the result states that

$$t_{period} \sim = \tilde{N}_f c_1^{\alpha/(n+1-\alpha)} c_2^{-1/(n+1-\alpha)} c_3^{1/(n+1-\alpha)}$$

$$\times \Psi_0^{(1+2\alpha)/[2(n+1-\alpha)]} V_p^{n/(n+1-\alpha)} q_{in}^{-1}, \tag{16}$$

where $\tilde{N}_f$ is a dimensionless constant, with a value of 1.44 for the parameters chosen here, in the limit of a large flow path length $L$ (section 4.1 of the supplementary material). This asymptotic formula is overlaid onto the numerically computed periods in figure 5; it should be clear that the asymptotic formula performs poorly for the relatively moderate values of $V_p$ used

here. This should not be a surprise since figure 4 demonstrates that, for the same values of $V_p$, the amplitude of oscillations is in fact sensitive to the inflow rate $q_{in}$.

    The same asymptotic solution also provides an estimate for the inflow rate $q_{in}$ at which zero effective pressure is first reached during the flood cycle, and our model ceases to be physically realistic as described above. The estimate is given by an analysis





of the flood initiation phase (section 4.4 of the supplementary material) as

$$q_{in} \sim \gamma_c c_1^{(n+1)/(\alpha n)} c_2^{-1/n} c_3^{(n+1)/(\alpha n)}$$

$$\times \quad \Psi_0^{3(n+1)/(2\alpha n)} (u_b h_r)^{-(\alpha-1)(n+1)/(\alpha n)} V_p, \tag{17}$$

This is superimposed on figure 6. The formula above is in general an underestimate of the value of $q_{in}$ at which zero effective pressures are reached, but clearly gives the correct scaling of how that value relates to $V_p$.

We are also able to construct a second asymptotic solution for the opposite case of a large water supply rate $q_{in}$ (section 5 of the supplementary material). This predicts rapid oscillations in the reservoir level (or effective pressure) whose amplitude slowly evolves to a steady value. These rapid oscillations however invariably involve effective pressure changing from negative to positive: at leading order the growth and shrinkage of the conduit is controlled by the creep 'closure' term, which becomes a creep opening term during about half of the cycle as in panel b of figure 8, and dissipative melting is a higher order correction

that ultimately dictates the amplitude that the rapid oscillations settle into over time. As discussed above, such solutions are clearly not physically viable, and the model must be amended to take account of ice-bed separation; the asymptotic solution here serves merely to confirm that negative effective pressures are a robust prediction of our model for these large inflow rates, when the possibility of ice-bed separation is not taken into account.

## 4    The spatially extended model

The analysis described above provides a comprehensive picture of the qualitative attributes of the lumped model (2). Here, we consider how well that lumped model represents the behaviour of the its more complete, spatially-extended counterpart (1). We begin by recreating the stability boundary diagrams of figure 1. The method by which the latter were computed explicitly (sections 3.1 and 3.2 of the supplementary material) cannot be applied directly to the extended model (1), since we have no closed-form solution to a linearized version of the model analogous to (12). Instead, we grid the $(q_{in}, \bar{V}_p)$ parameter space

used in figure 1 finely. For each $(q_{in}, \bar{V}_p)$ pair, we discretize (1), compute steady states and perform a linear stability analysis numerically as described in Schoof et al. (2014); this allows us to delineate regions of instability, although in a less sophisticated way.

    Results are shown in figure 7. It is clear that the lumped model consistently underestimates the range of parameter values over which steady drainage is unstable. The onset of instability at the transition from cavity- to channel-like conduit behaviour

appears to remain robust except in the case of small domain lengths $L$ (panel c), for which the system appears to be unstable for combinations of small storage capacities $\bar{V}_p$ and inflow rates $q_{in}$. Where the lumped model typically underestimates instability is at low storage capacities, and at large flow path lengths $L$. The latter is particularly significant since we have previously attributed stabilization to the effect incipient reservoir drainage reducing the hydraulic gradient along the flow path and therefore reducing flow through the conduit.

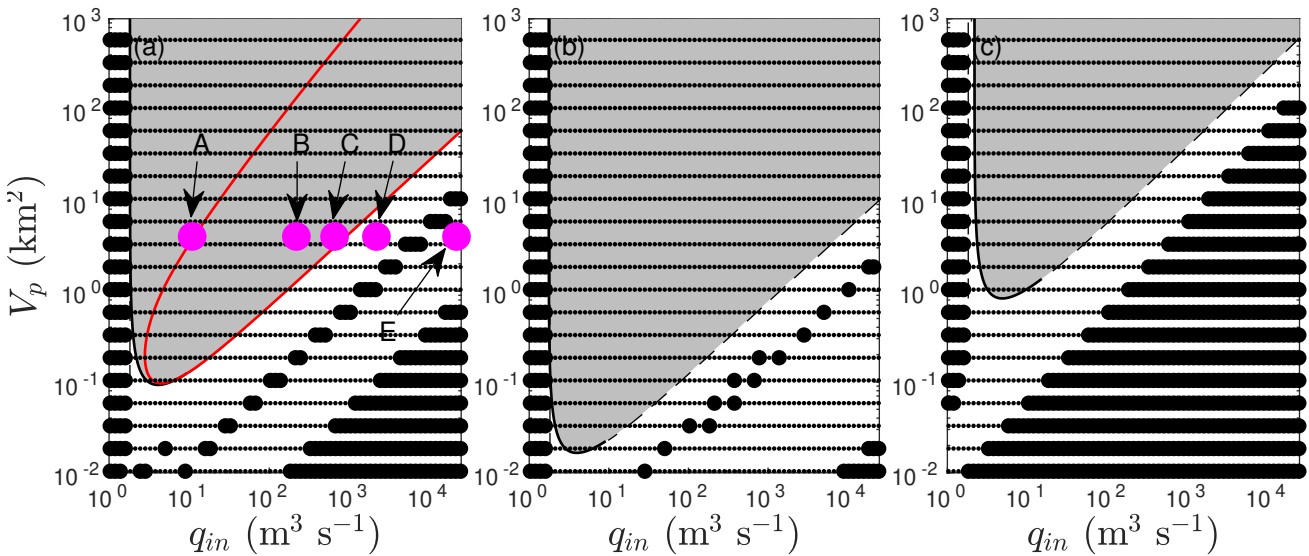

**Figure 7.** The stability boundaries of figure 1 with $L$ =50 km (a), 250 km (b) 10 km (c) plotted in black; unstable regions in the lumped model (2) are marked in grey. Solid black circles indicate parameter combinations for which the steady state solution to the full model (1) is stable, small dots indicate that the full model is unstable. The magenta markers in panel (a) indicate parameter values for which the evolution of the full, spatially extended model has been computed up to finite amplitude (figures 8–10). The red curve in panel (a) is the same boundary as in figure 6 indicating where negative effective pressures are encountered in the lumped model.

While stabilization at large water input rates $q_{in}$ does eventually occur, this occurs at values that can be several orders of magnitude larger than those predicted by the lumped model, especially for the case of a large flow path lengths $L$. Furthermore, for a given storage capacity $\bar{V}_p$, the lumped model predicts that there is a single interval of inflow rate values $q_{in}$ over which instability occurs. The spatially extended model by contrast has two or more such intervals for most values of $\bar{V}_p$, with a narrow

5   region of stability between (the diagonal bands of solid black diamonds in panels a and b of figure 7).

To understand this discrepancy better. we have solved for the nonlinear evolution of the draiange system as described by the spatially extended model (1) for the parameter values indicated by magenta circles in panel a of figure 7.

The there smallest values of $q_{in}$ all correspond to unstable steady states in the lumped model (2), and we can compare spatially extended and lumped solutions. Figure 8 shows results. While the spatially extended solution is an infinite-dimensional

10   dynamical system and strictly speaking cannot be visualized using a phase plane, we can overlay plots of conduit size $S(0,t)$ at the upstream end of the conduit against effective pressure $N(0,t)$ at the same location onto the $S-N$ phase plot for the lumped model. This is shown in the top row of panels. In all three cases, the extended models settles into a limit cycle, and for small and intermediate $q_{in}$, we find good agreement between extended and lumped models; this only breaks down as the upper critical value of $q_{in}$ at which the lumped model stabilizes is approached.

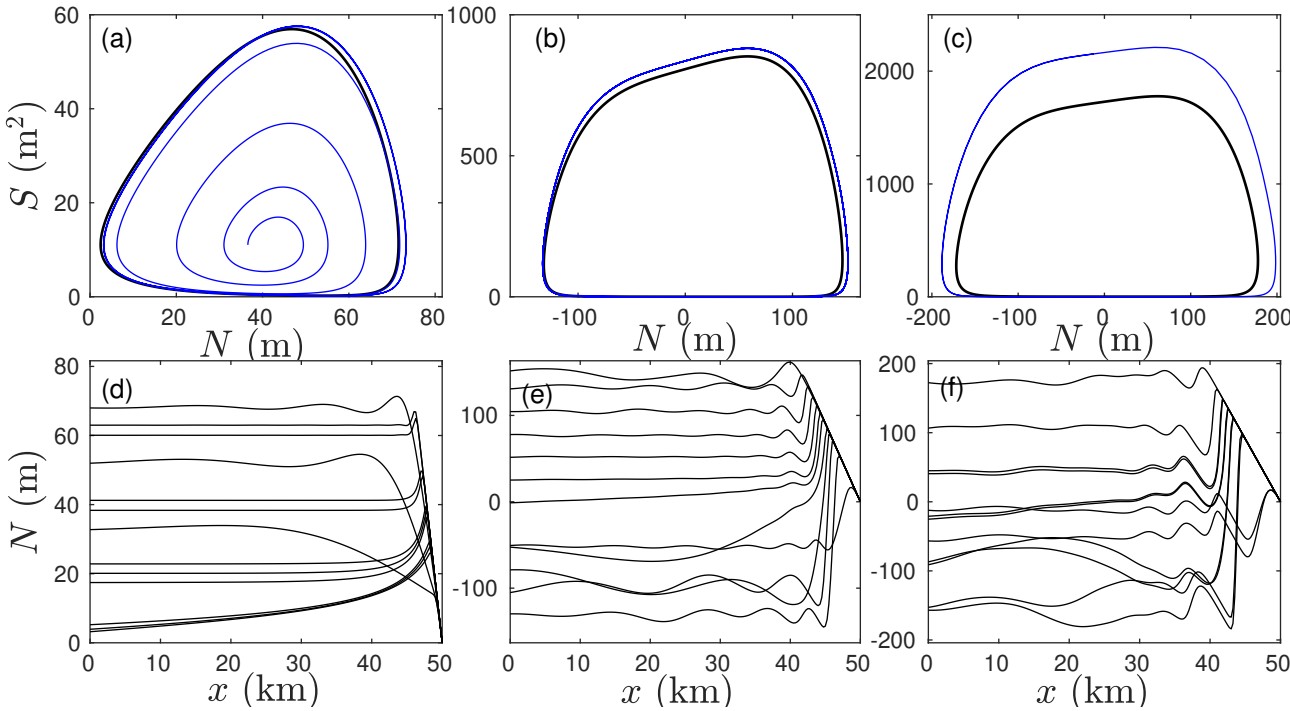

**Figure 8.** Each column represents one of the magenta markers in figure 7a. For each column, $V_p = 4$ km$^2$ and $L = 50$ km, while $q_{in} = 10.9$ m$^3$ s$^{-1}$ (panels a, d, marker A in figure 7a.), $218.4$ m$^3$ s$^{-1}$ (panels b, e, marker B), $655$ m$^3$ s$^{-1}$ (panels c, f, marker C). The top row (panels a-c) displays phase planes of $(N(0,t), S(0,t))$ for the full model (1) (blue) superimposed on the phase plots for the lumped model (2) (black). The bottom row shows snapshots of $N(x,t)$ against $x$ for periodic solutions of the full model at intervals of 93 days (panel d) 5.5 days (e) and 4.6 days (f).

The lower row of panels in figure 8 shows snapshots of $N(x,t)$ against $x$ for the correspnding limit cycle shown in the top panel of the same column. For the smaller two values of $q_{in}$, we see that pressure gradients $\partial N / \partial x$ are in general moderate away from the glacier terminus and therefore do not contribute significantly to the hydraulic gradient $\Psi$: this was the basis for the reduction of the spatially extended model (1) to the lumped form (2), and explains the good agreement. For larger $q_{in}$,
5  effective pressure $N(x,t)$ along the conduit starts to develop wave-like structures, causing the approximation of negligible pressure gradients to break down, and the discrepancy between lumped and extended model grows.

The extended model remains unstable beyond the stability boundary model of the lumped model, but our numerical solutions no longer support the conclusion that the system necessarily settles into a limit cycle. Figure 9 shows the evolution of the system for $V_p = 408$ m$^3$ Pa$^{-1}$ s$^{-1}$ (a lake with a surface area of 4 km$^2$) and $q_{in} = 2.18 \times 10^3$ m$^3$ s$^{-1}$ as in the magenta marker labelled
10  'D' in figure 7a (the fact that this is an inflow rate of biblical proportions is not as relevant as it may at first seem, as we discuss in section 5 below). Panels a and b show the growth of rapid oscillations in the maxima of $N$ and $S$ along the flow

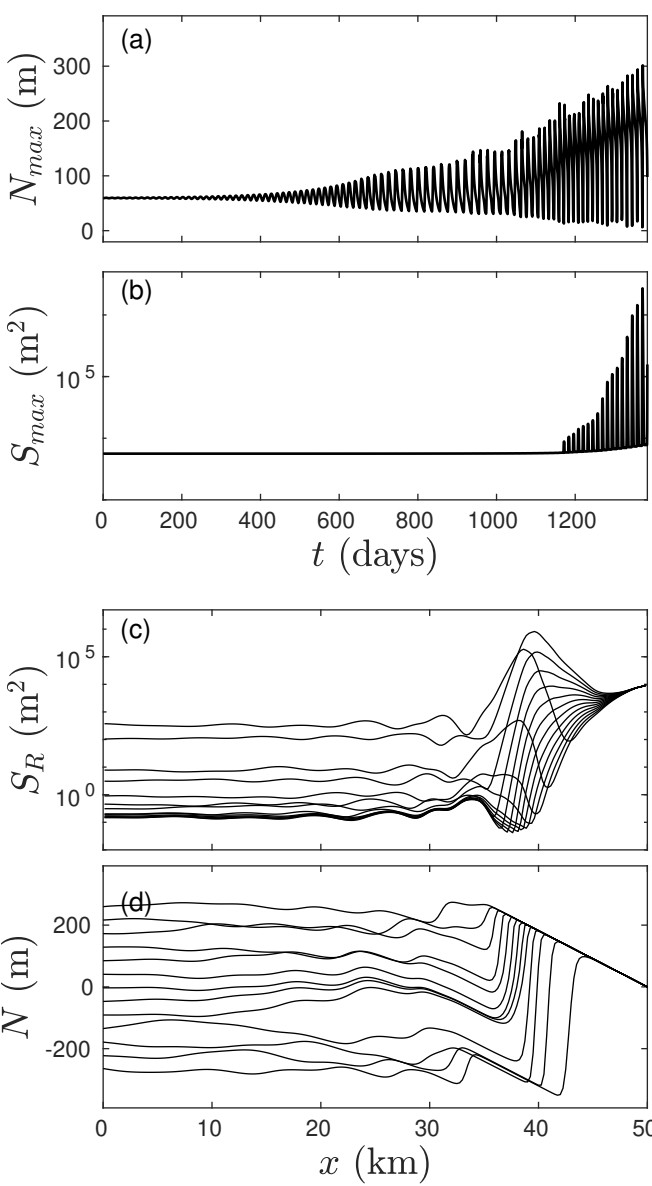

**Figure 9.** Finite-amplitude evolution of the instability for the magenta dot marked 'D' in figure 7a. (a): the maximum of $N(x,t)$ along the flow path plotted against time $t$. (b): the maximum of $S(x,t)$ with respect to $x$ plotted against $t$ (c): snapshots of $S(x,t)$ at intervals of 0.93 days towards the end of the simulation (d): snapshots of $N(x,t)$ at the same points in time as in (c). Note the logarithmic vertical scale in (c).



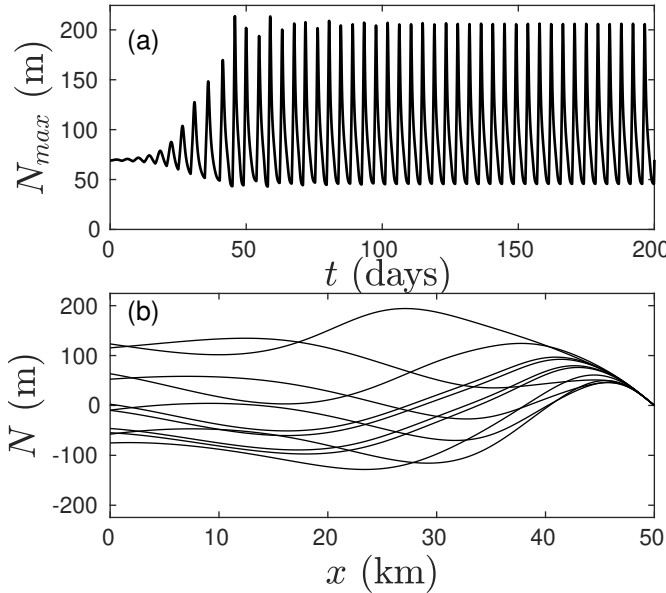

**Figure 10.** Finite-amplitude evolution of the instability for the magenta dot marked 'E' in figure 7a. (a): the maximum of $N(x,t)$ along the flow path plotted against time $t$. (b): snapshots of $N(x,t)$ at the same points in time as in (c).

path over time, but without the growth becoming bounded. Panels c and d show snapshots over the last two oscillations in the computation. Note that the vertical scale in panel c is logarithmic. What we see is that conduit size develops an aneurysm-like, massively enlarged feature near the terminus, blocked by a narrow constriction that requires an extremely large pressure gradient to overcome (note that panel c uses a logarithmic verticla scale) . The simulation eventually terminates when the solver

fails to converge, and it is unclear whether this is merely a computational problem or indicates that the true continuum solution becomes pathological or ceases to exist.

For even larger $q_{in}$, a narrow band of stable values is passed and a different instability ensues as shown in figure 10. This instability appears to lead to bounded growth, again marked by rapid oscillations whose amplitude slowly grows, and significant pressure gradients along the flow path.

Once again, a key observation is that the spatially extended model predicts negative effective pressures long before any kind of limit cycle is reached for all but one of the solutions shown, and that the discrepancies between lumped and spatially extended models occur only where this is the case. Moreover, they occur for water supply rates to the reservoir that far exceed values that would be plausible for typical glacier-dammed lake systems: the lumped model appears to be robust for the latter, including its prediction of where overpressurization of the drainage system with negative $N$ first occurs.

Nonetheless, as we will discuss shortly, this does not render some of the more exotic instabilities shown in figures 8–10 irrelevant: The parameter values we have chosen here were deliberately chosen to reflect typically large reservoirs dammed by





low-angled and large glaciers. Much smaller reservoirs in shorter more steeply-angled glaciers are liable to give rise to some of the instabilities that lie outside the reach of standard glacier-dammed lakes.

## 5 Discussion

### 5.1 Glacier-dammed lakes and small reservoirs

We have shown that a drainage system that switches between cavity-like and channel-like behaviour spontaneously is capable of supporting periodic outburst floods from a glacier-dammed reservoir. At issue here is the initiation of the flood: as discussed by Ng (1998) and Fowler (1999), if the conduit consists purely of a Röthlisberger-type channel, kept open only by melting due to heat dissipated in the turbulent flow of water through the channel, then the initiation of floods can be delayed more and more, without a limit cycle emerging. Both of these authors proposed that englacially-routed water supplied to the channel

should keep it open at a minimum size, and the migration of the flow divide within the channel as the lake fills then becomes the control on flood initiation. Our theory differs from theirs in the sense that no englacial water supply is necessary and the lake can continuously leak water through a linked-cavity-type drainage system that spontaneously becomes 'channel-like' and undergoes runaway enlargement through dissipation-driven melting as the lake fills.

At the heart of the oscillatory behaviour of glacier-dammed lakes is exactly that instability of a channel, which prevents

steady discharge of the water supplied to the reservoir (Nye, 1976; Ng, 1998; Fowler, 1999). As described in Schoof et al. (2014) and section 3.1 above, this instability does not occur when the conduit draining the reservoir acts as a set of linked cavities: when these are capable of steadily draining the water supplied to the reservoir, no outburst floods occur.

By contrast, for typical glacier-dammed lakes, in which moderate inflows $q_{in}$ exceed the drainage capacity of a linked cavity system but the the large storage capacity $V_p$ ensures that the lake fills slowly compared with the time scale over which a basal

conduit can evolve, the cycle of filling and draining follows the characteristic sequence of a long filling period in which outflow from the lake is negligible, followed by a brief onset period in which the conduit starts to experience significant melt-driven enlargement but the lake level continues to rise, and an even shorter outburst flood in which inflow to the lake is dwarfed by drainage through the subglacial conduit. For a given lake size $V_p$, larger inflow rates will slowly increase the amplitude of the lake level fluctuation during the flood cycle (figure 4), while significantly shortening the length of the flood cycle (figure 5).

As in Ng (1998) and Fowler (1999), our flood initiation mechanism can also be too slow to respond to water input and lead to our models (1) and (2) predicting negative effective pressures in the reservoir: in that case, the glacier ought to float and flood initiation is likely to take the form a sheet flow that subsequently channelizes, as explored in Schoof et al. (2012) and Hewitt et al. (2012). This effect is not included in our models here, but we are at least able to give an asymptotically valid (in the limit of very large reservoir sizes $V_p$) criterion for the water supply rate at which the change in flood initiation between

unstable conduit growth and partial glacier flotation occurs (equation (17)). Adapting our model to account for this alternative flood initiation mechanism is the obvious next step to take.

We have also investigated a mechanism previously identified in Schoof et al. (2014), by which flow out of a reservoir can be stabilized when the storage capacity in the reservoir is relatively small, so that typical drainage rates (comparable to the inflow

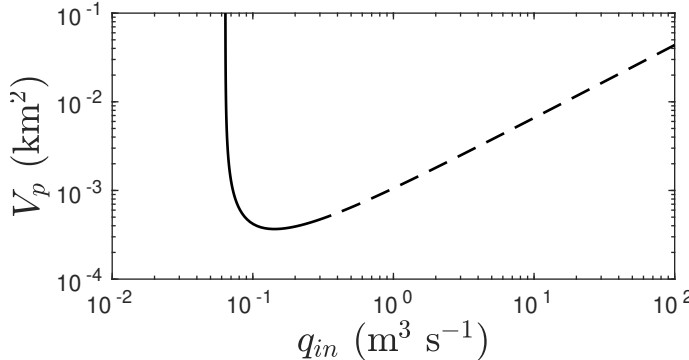

**Figure 11.** Stability boundary plotted in the same way as in figure 1, but with $\Psi_0 = 1630$ Pa m$^{-1}$, $u_b h_r = 1.05 \times 10^{-07}$ m$^2$ s$^{-1}$, and we put $L = 5000$ m.

rate $q_{in}$) lead to adjustment of effective pressure in the lake much faster than the conduit can evolve due to wall melting. The rapid response of reservoir water levels can have a large enough effect on hydraulic gradients to stabilize the flow. This is true at least for a short enough drainage system. Importantly, this happens at water throughput rates that are completely unrealistic for typical, large glacier lakes (and even these rates are underestimated by a simplified, 'lumped' model as described in section 4).

The reason why this stabilization mechanism is relevant is that it may explain why much smaller water reservoirs that are typically not recognized as lakes but nonetheless provide storage capacity (such as large moulins) do not invariably generate outburst-flood type behaviour: the stability diagram of figure 1 was generated for parameter values chosen to represent large lakes dammed by a long glacier with a small surface slope, but can easily be rescaled to represent other glacier geometries, reservoir storage capacities and water throughput. The relevant scaling is explored in section 2 of the supplementary material.

Here, we confine ourselves to a simple numerical demonstration: figure 11 recalculates the stability boundary for steeper, shorter valley glacier with a relatively small reservoir sizes. We use the same parameter values as in table 1, except for $\Psi_0$, $u_b h_r$ and $L$. The 'upper' stability boundary (the larger critical value of $q_{in}$ where the lumped model (2) becomes stable again) is much more accessible for reasonable flow rates, and some of the more exotic manifestations of instability explored in section 2.1 are more likely to be within reach of typical water input rates.

With this as motivation, we also briefly discuss a second instability. This diverges from the classical picture of a glacier outburst flood in that it does not require channel-like behaviour but can occur in a purely cavity-based drainage system. We analyze this instability briefly below, since the work already done in the present paper provides the necessary framework.

### 5.2    Distributed storage: a modification of Nye's instability

In section 2, we formulated a model for a single reservoir, for which Nye's instability predicts the onset of oscillations when the conduit becomes channel-like. In Schoof et al. (2014, section 5), the case of spatially spread-out storage capacity, for





instance in the form of numerous basal crevasses arrayed along the flow path, was considered in addition to a single reservoir, A numerical linear stability analysis was used to demonstrate that instability can also occur for cavity-like conduits (which are generally stable for a single reservoir) for the case of such distributed water storage. As in the case of a single reservoir, Schoof et al. (2012) find that there is a finite range of values $q_{in}$ for which instability then occurs, only that this extends to lower values of $q_{in}$, where the conduit is cavity-like.

Here we build on the analysis in section 3.1 to shed further light on their resultts. The model used in Schoof et al. (2012) was

$$\frac{\partial v(N)}{\partial t} + \frac{\partial q}{\partial x} = 0, \tag{18a}$$

$$\frac{\partial S}{\partial t} = c_1 q\Psi + v_o(S) - v_c(S, N), \tag{18b}$$

$$q = q(S, \Psi), \tag{18c}$$

$$\Psi = \Psi_0 + \frac{\partial N}{\partial x}, \tag{18d}$$

where $v(N)$ is a *decreasing* function of $N$ that describes storage of water per unit length of the conduit, and the rest of the notation used replicates that of the single-reservoir model (1). Below, we will denote by

$$v_p = -\frac{\mathrm{d}v}{\mathrm{d}N}, \tag{19}$$

the equivalent of $V_p$ in (1). This model also requires boundary conditions; an obvious choice is a prescribed flux $q = q_{in}$ at the inflow $x = 0$, with no actual reservoir there, and again $N = 0$ at the terminus $x = L$.

In order to take advantage of the work already done in section 3.1, we adopt an analytical approach to understanding the instabilities of this modified drainage model, complementing the numerical stability analysis of Schoof et al. (2014) and providing additional insight into the feedbacks involved.

With a finite domain size and the proposed boundary conditions above, a steady state solution to (18) will in general have spatial structure as in Schoof et al. (2014), and a linearization around that steady state will lead to a boundary value problem with non-constant coefficients that is not amenable to a closed-form solution. To avoid this, we concentrate here on shorter length scales, and assume that we can use (18) with periodic boundary conditions at these scales. This yields a spatially uniform steady state solution defined implicitly by

$$c_1 \bar{q}\bar{\Psi} + v_o(\bar{S}) - v_c(\bar{S}, \bar{N}), \tag{20a}$$

$$\bar{q} = q(\bar{S}, \bar{\Psi}), \tag{20b}$$

$$\bar{\Psi} = \Psi_0. \tag{20c}$$

with a constraint on $\bar{N}$ through the prescribed volume of water in the system (which is preserved due to the assumed periodic boundary conditions). Note that this is closely analogous to (8) but simpler, as the hydraulic gradient here does not contain the gradient term retained in (8c).





Linearizing as $N = \bar{N} + N' \exp(ikx + \lambda t)$, $S = \bar{S} + S' \exp(ikx + \lambda t)$, we find

$$-\bar{v}_p \lambda N' + ikq_S S' - k^2 q_\Psi N' = 0, \tag{21a}$$

$$\lambda S' - (c_1 q_S \bar{\Psi} + v_{o,S} S' - v_{c,S}) S'$$
$$- \left[ ikc_1(q_\Psi \bar{\Psi} + \bar{q}) - v_{c,N} \right] N' = 0. \tag{21b}$$

The eigenvalue $\lambda$ satisfies a quadratic of the form (10), with solution (12), but now with $a_1$ and $a_2$ given by

$$a_1 = (c_1 q_S \bar{\Psi} + v_{o,S} - v_{c,S}) - \bar{v}_p^{-1} k^2 q_\Psi \tag{22a}$$

$$a_2 = \bar{v}_p^{-1} \left\{ k^2 \left[ c_1 q_S \bar{q} + q_\Psi (v_{c,S} - v_{o,S}) \right] + ikq_S v_{c,N} \right\} \tag{22b}$$

The only term that differs intrinsically from (11) is $a_2$, which now has an imaginary part. We see that the real part of $a_2$ is still invariable positive, so the real part of $a_1 - 4a_2$ is smaller than $a_1$. However, we can no longer conclude that the real part of $\lambda$

has the same sign as $a_1$ except in appropriate limits.

For short wavelengths (large $k$), the system is stable with $\Re(\lambda) < 0$: the two limiting forms of the two eigenvalues are

$$\lambda_1 \sim -\bar{v}_p q_\Psi k^2, \qquad \lambda_2 \sim -\frac{c_1 q_S q + q_\Psi (v_{c,S} - v_{o,S})}{q_\psi},$$

and the assumptions about $q_\psi$, $q_S$, $v_{o,S}$ and $v_{c,S}$ in (6)–(7) ensure that both of these expressions are negative. Limited storage over short length scales prevents large variations in water discharge and prevents short-wavelength disturbances in conduit size and effective pressure from growing.

Conversely, at sufficiently large wavelengths (small $k$), we have $a_1 \sim (c_1 q_S + v_{o,S} - v_{c,S})$, $a_2 \sim O(k)$, and a channel-like

conduit will be unstable with one eigenvalue behaving as $\lambda \sim (c_1 q_S + v_{o,S} - v_{c,S})$, while a cavity-like conduit is stable at long wavelengths. This is simply Nye's instability at work. Clearly, then, there is a critical wavelength at which a channel-like conduit with distributed water storage will become unstable (since it is stable at short wavelengths and unstable at long wavelengths), and the critical wavelength is determined by the various parameters in the model. How large that wavelength is matters, because we have assumed in setting up our stability analysis that we are looking at only a relatively small part of

the domain, with periodic boundary conditions. If the critical wavelength approaches the full domain size $L$, our simplified analysis breaks down, and the instability is no longer guaranteed to occur. This is consistent with figure 7 of Schoof et al. (2014), where channel-like conduits eventually become stable for large enough water supply rates.

Notably, the critical wavelength for instability of channel-like conduits will depend on storage capacity $\bar{v}_p$. The more storage capacity there is, the smaller all the terms containing $k$ are in (22) for a given wavenumber $k$, and the more likely the first term

$(c_1 q_S + v_{o,S} - v_{c,S})$ in the definition of $a_1$ is to dominate the eigenvalue $\lambda$, leading to instability for a channel-like system. To increase the size of the potentially stabilizing terms therefore requires larger wavenumbers $k$, and therefore a larger range of short wavelengths is likely to be unstable. Although we are not able to address a system of finite length directly with this approach, our results indicate that systems of limited size may remain stable if storage capacity is sufficiently limited, and this is confirmed by Schoof et al. (2014).





However, an interesting possibility is left open by (22). Even if the conduit is cavity-like with $(c_1 q_S + v_{o,S} - v_{c,S}) < 0$ and therefore $a_1 < 0$, it may still be possible to have an eigenvalue with positive real part and hence instability. This is the case because $a_2$ has a non-zero imaginary part.

One particular case in which an instability due to this term may occur is with limited storage capacity (so $\bar{v}_p^{-1}$ is large) and at an intermediate wavelengths range (so $k$ is small but not too small). To be definite in identifying that parameter regime, suppose we scale the model; with the constitutive relations (1g), this is done by defining scales $[S]$, $[N]$, $[q]$, $[x]$ and $[t]$ through $c_3[S]^\alpha[\Psi]^{1/2} = [q]$, $[S]/[t] = c_1 c_3 [S]^\alpha[\Psi]^{3/2} = c_2[S][N]^n$, $[\Psi] = [N]/[x] = \Psi_0$ and a constraint on $v([N])$ through the prescribed quantity of water in the system. Dimensionless variables are then $N^* = [N]^{-1}N$, $S^* = [S]^{-1}S$, $q^* = [q]^{-1}q$, $x^* = [x]^{-1}x$, $t^* = [t]^{-1}t$. Dropping asterisks, the model then becomes (18) with (1g) but with periodic boundary conditions, and with $c_1 = c_2 = c_3 = 1$ and $\Psi_0 = 1$. The stability analysis applies as stated, and the derivatives $q_S$, $q_\Psi$, $v_{c,S}$, $v_{c,N}$ and $v_{o,S}$ are $O(1)$ quantities. $\bar{v}_p$ is replaced by a dimensionless counterpart

$$\tilde{v}_p = v_p c_1^{(n+2)/n} c_2^{-2/n} c_3^{(2-n)/\alpha} \bar{q}^{\prime 2(\alpha-1)-n]/\alpha|}$$

$$\times \Psi_0^{(2\alpha m + 4\alpha + 2 - n)/(2\alpha n)},$$

and $k$ as well as $\lambda$ are scaled wavenumbers and growth rates.

We can make a cavity-like conduit unstable provided $k \ll 1$ and $\tilde{v}_p \ll k^3$. Then $a_2 \sim i\tilde{v}_p^{-1} k q_S v_{c,N}$ and $a_1^2 - 4a_2 \sim [(q_S + v_{o,S} - v_{c,S}) - \tilde{v}_p^{-1} k^2 q_\Psi]^2 - i4\tilde{v}_p^{-1} k q_S v_{c,N}$. From our constraints on $\tilde{v}_p$ and $k$, $k \ll 1$ and $\tilde{v}_p^{-1} k \gg \tilde{v}_p^{-2} k^4$ so

$$a_1^2 - 4a_2 \sim -4i\tilde{v}_p^{-1} k q_S v_{c,N}.$$

5   Then we have

$$\lambda = \frac{1}{2}\left(a_1 \pm \sqrt{a_1^2 - 4a_2}\right)$$

$$\sim \frac{1}{2}\left[(q_S + v_{o,S} - v_{c,S}) - \tilde{v}_p^{-1} k^2 q_\Psi \pm 2i\frac{1+i}{\sqrt{2}}\sqrt{\tilde{v}_p^{-1} k q_S v_{c,N}}\right]$$

But with our assumptions on $k$ and $\tilde{v}_p$, this is

$$\lambda \sim \pm(1-i)\sqrt{\frac{\tilde{v}_p^{-1} k q_S v_{c,N}}{2}} \qquad (23)$$

10   Choosing the $+$ sign ensures an eigenvalue with positive real part. Importantly, this instability corresponds to a growing wave that propagates, in this case downstream as the imaginary part of $\lambda$ is then negative. It is also of the same size as the real part, so propagation is not slow. This unstable wave is not the result of Nye's instability as the conduit is cavity-like. Instead, it is the result of an interaction between the dependence of the conduit closing rate on effective pressure and the dependence of water drainage (which affects effective pressure through water storage) on conduit size: the dominant balance that underpins it is

$$\frac{\partial S'}{\partial t} \sim v_{c,N} N', \qquad (24a)$$

$$-\bar{v}_p \frac{\partial N'}{\partial t} + q_S \frac{\partial S'}{\partial x} \sim 0, \qquad (24b)$$





where the perturbed conduit evolution does not include the melting term $c_1 q \Psi$ at all. Equations (24) can be combined into the single 'diffusion' equation $\bar{v}_p \partial^2 N'/\partial t^2 - q_s v_{c,N}^{-1} \partial N'/\partial x \sim 0$, except that the roles of time $t$ and space $x$ have been reversed. We can identify the positive feedback causing growth as a the result of a phase lag, where $S'$ leads $N'$ in phase and thus ensures that $\partial S'/\partial x$ is positive where $N'$ has a maximum, therefore ensuring that $\partial N'/\partial t$ is also positive.

Naturally, this sketch is incomplete, as the 'diffusion' problem with the roles of $x$ and $t$ reversed is not well-posed. The growth rate in (23) grows unboundedly as wavelength $k^{-1}$ approaches zero, which is a clear sign something is amiss. The discussion above was merely designed to identify a positive feedback that can drive growth; a negative feedback is necessary to ensure there is a fastest growing wavelength and that short wavelengths are damped as already discussed.

## 6    Conclusions

As demonstrated previously using a much more restricted sweep of parameter space in Schoof et al. (2014), we have shown that drainage systems capable of switching spontaneously between channel- and cavity-like behaviour are stable in the presence of a localized water reservoir at low and high water throughput, with an unstable intermediate range of water fluxes. In that range, spontaneous oscillations in reservoir level will occur, driven by Nye's (1976) instability mechanism driving outburst floods. These outburst floods turn out to be regular, periodic oscillations in water level at least at low-to-moderate water input, where
a simplified, 'lumped' model of reservoir drainage generally reproduces the results of a more sophisticated, spatially-extended drainage model. At high water throughput rates, our results are more equivocal as to the emergence of such limit cycles in the model used; in any case, the model necessarily breaks down physically (as opposed to mathematically) because it predicts that the oscillations will invariably reach negative effective pressures and therefore flotation of the ice dam at such large throughput rates.

It is worth pointing out that part of our focus has been on the emergence of limit cycle solutions in order to identify how the flood initiation mechanism can prevent flood magnitude from increasing progressively from cycle to cycle as observed in Ng (1998) and Fowler (1999), and to present an alternative mechanism for suppressing the continued growth in amplitude from that considered by Fowler (1999), whose lake is necessarily 'sealed' between floods. That mechanism is the ability of a cavity-like drainage system that remains during the reservoir recharge phase to switch to a channlized drainage mode.

The fact that we illustrate this by showing evolution towards a limit cycle is not at odds with the chaotic behaviour observed by Kingslake (2015) using a variant of Fowler's (1999) model: the chaotic behaviour is intrinsically the result of a time-varying water input to the reservoir, which we have not studied in this paper. It is entirely possible, and in fact likely, that our model will also behave chaotically under such time-varying water input rates $q_{in}(t)$. The point of demonstating that limit cycles emerge was rather to underline that the growth of Nye's (1976) instability is bounded in our model without having to appeal to
additional physics.

The lower cut-off to the drainage instability that leads to outburst floods corresponds to the drainage system switching to a cavity-like state under steady flow conditions when water input to the reservoir can be drained steadily by those cavities. The mechanism for the cut-off at high water throughput rates is harder to identify. The lumped model predicts that the high





sensitivity of water level in the reservoir to the evolution of the draining conduit will induce water pressure gradients that reduce flow as the conduit grows, and therefore suppress its enlargement due to heat dissipation. The threshold at which this happens is however significantly underestimated by lumped model relatively to its spatially-extended counterpart, which develops more wave-like instabilities at higher water throughput rates.

In closing, we have also investigated how wave-like instabilities can occur when the water reservoir is not localized but spread out or 'distributed' along the flow path (for instance, in the form of many small reservoirs like basal crevasses). This type of instability was first observed in Schoof et al. (2014), and persists even where water throughput is insufficient to lead to channel-like behaviour. Adapting the stability analysis performed on a model with localized storage, we have shown that Nye's instability persists, but also that a second instability mechanism emerges, in which a phase shift between water storage

and flux arises that causes water to accumulate in regions where effective pressure is already low.

     Future work is likely to focus on capturing the role of overpressurization of the drainage system in initiating and mediating the instability driving outburst floods, since flood initiation at water pressures below flotation is confined to a relatively small part of parameter space, and the model predicts that reaching zero effective pressure and initiation by partial flotation of the ice dam is likely to be common.

**7   Code availability**

The MATLAB code used in the computations reported is included in the supplementary material.

*Acknowledgements.*   Discussions with Rob Vogt, Ian Hewitt and Garry Clarke are gratefully acknowledged. This work was supported by an NSERC Discovery Grant.



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



## Appendix A: Asymptotic solutions for large lakes with moderate inflow

This appendix provides only a brief sketch of the derivation of an asymptotic solution for a limit cycle solution in the case where the reservoir volume is large and inflow is sufficient to ensure that the reservoir cannot be drained by a cavity-like conduit, but also that the conduit size evolves much faster than the time scale over which the reservoir fills. Full details are

given in section 4 of the supplementary material.

The solution is developed from a parameteric limit of the model (2) with (1g), where we assume $S_0 = \infty$. We scale as $N^{**} = N/[N]'$, $S^{**} = S/[S]'$, $t^{**} = t/[t]'$, where the scales are defined through $[S]'/[t]' = c_1 c_3 [S]'^\alpha \Psi_0^{3/2} = c_2 [S]'[N]'^n$ and $V_p [N]'/[t]' = c_1 [S]'^\alpha \Psi_0^{1/2}$. Substituting and omitting the asterisks immediately, we obtain

$$\dot{S} = S^\alpha |1 - \nu N|^{3/2} + \delta - S|N|^{n-1}N \tag{A1a}$$

$$\dot{N} = -\epsilon + S^\alpha |1 - \nu N|^{-1/2}(1 - \nu N) \tag{A1b}$$

where $\nu = [N]'/(\Psi_0 L)$, $\delta = u_b h_r/(c_2 [S]'[N]'^n)$ and $\epsilon = q_{in}/(c_1 [S]'^\alpha \Psi_0^{1/2})$.

We assume that the exponents $n$ and $\alpha$ satisfy $n + 1 > \alpha > 1$, in which case the asymptotic solution we develop is valid when

$$\delta \ll \epsilon \lesssim \delta^{(\alpha-1)(n+1)/(\alpha n)} \ll 1$$

and $\nu \lesssim 1$.

The main flood phase is described by omitting terms of $O(\delta)$ and $O(\epsilon)$

$$\dot{S} = S^\alpha |1 - \nu N|^{3/2} - S|N|^{n-1}N, \tag{A2a}$$

$$\dot{N} = S^\alpha |1 - \nu N|^{-1/2}(1 - \nu N), \tag{A2b}$$

where matching with the initiation phase described later requires that $(N, S) \to (0, 0)$ as $t \to -\infty$. The transformation $P = N/S$ allows the system (A2) to be re-cast in such a way as to demonstrate that the orbit into the fixed point $(0, 0)$ is unique, so there is only one flood phase solution. The orbit terminates at a finite $N = \tilde{N}_f$ as $t \to \infty$; this then sets the amplitude of the lake level fluctuations that give the asymptotic formula for the flood cycle period (16).

The refilling phase is described by the rescaling $\tilde{N} = N$, $\tilde{S} = \delta^{-1}S$ and $\tilde{t} = \epsilon(t - t_f)$, where $t_f$ is the time of the last flood. At leading order,

$$0 = 1 - \tilde{S}|\tilde{N}|^{n-1}\tilde{N}, \tag{A3a}$$

$$\frac{\mathrm{d}\tilde{N}}{\mathrm{d}\tilde{t}} = -1; \tag{A3b}$$

conduit size is quasi-steady and cavity-like, while lake level evolves purely because of inflow.

The refilling phase ends as $N \to 0$ and cavity size becomes large. The relevant rescaling becomes

$$\hat{N} = \delta^{-(\alpha-1)/(\alpha n)}\tilde{N}, \qquad \hat{S} = \delta^{(\alpha-1)/\alpha}\tilde{S},$$





$$\hat{t} = \delta^{-(\alpha-1)/(\alpha n)}(\tilde{t} - N_f)$$

and gives at leading order

$$\epsilon \delta^{-(\alpha-1)(n+1)/(\alpha n)} \frac{\mathrm{d}\hat{S}}{\mathrm{d}\hat{t}} = \hat{S}^\alpha + 1 - \hat{S}|\hat{N}|^{n-1}\hat{N}, \tag{A4a}$$

$$\frac{\mathrm{d}\hat{N}}{\mathrm{d}\hat{t}} = -1, \tag{A4b}$$

where we have assumed $\epsilon \sim \delta^{(\alpha-1)(n+1)/(\alpha n)}$; the alternative case of $\epsilon \sim \delta$ is described in the supplementary material. The key

5 to (A4) is that it predicts finite-time blow-up in $\hat{S}$ at some finite $\hat{N} = \hat{N}_c$ that depends purely on the value of $\epsilon \delta^{-(\alpha-1)(n+1)/(\alpha n)}$. This is the smallest value of effective pressure (rescaled, of course) that is reached during the drainage cycle. The larger $\epsilon \sim \delta^{(\alpha-1)(n+1)/(\alpha n)}$, the smaller and eventually more negative $\hat{N}_c$ becomes; there is therefore a critical value of $\epsilon \sim \delta^{(\alpha-1)(n+1)/(\alpha n)}$ at which $\hat{N}_c = 0$. When rendered in dimensional form, this value gives the formula (17).