# Peer review of "An analysis of instabilities and limit cycles in glacier-dammed reservoirs"

_The Cryosphere, 2019_

## Referee Comment (RC1) · Anonymous Referee #1 · 29 Aug 2019

**An analysis of instabilities and limit cycles in glacier-dammed reservoirs**

CHRISTIAN SCHOOF

THE CRYOSPHERE TC-2019-138

*Referee's report*

This is an awkward paper to referee. Being Christian Schoof, there is not going to be anything technically wrong with it, but from my perspective it is not properly thought out.

Nor is it well presented. The very first thing to say is that the text is littered with typographical and grammatical errors, so many that I will not list them all here. But there are such errors on page 1, line 16; 1,24; 2,5; 2,6; 2,13; 2,15; 2,20 (twice); 2,29 (the whole second half); 2,35; and so on, and on, and on. Perhaps the best is saved for last, where Schoof's own 2012 paper is mis-referenced (it is part 1, not part 2). Incidentally, all the poor authors cited in the references are demoted to a single initial each.

The paper concerns a model (which is analysed in both a 'lumped' form and a spatially dependent one) for subglacial floods, or jökulhlaups. The paper is motivated heavily by previous work of Fowler and Ng, and seeks to modify this earlier work, by allowing for the case where there is no 'seal' of the subglacial (or ice-dammed) lake, which can then continually leak between floods, as is the case for Summit Lake, according to Fisher in 1973. Note: Fisher, not Fischer.

The improvement consists of showing that with an extra term in the closure equation of the classical Nye-Röthlisberger theory, the model will describe limit cycles even in the absence of a seal. This seems to me the principal achievement of this paper. The extra term invoked is an ingenious addition due to Schoof in 2010 which allows the description of both cavity drainage and channel drainage within the confines of a single model. It is worth offering some comments on this addition.

In its original form, the extra term appears as the first term on the right hand side of the closure equation

$$\dot{S} = u_b h_r + c_1 \Psi Q - c_2 N^n \left( \frac{Q}{c_3 \sqrt{\Psi}} \right)^{0.8},$$

and it describes the opening of cavities by ice flow (velocity $u_b$) over bedrock bumps (height $h_r$). The steady state of this equation provides for both channels ($N$ increases with $Q$) and linked cavities ($N$ decreases as $Q$ increases). There are several comments to make.

(i) We might suppose in reality that $u_b$ will itself depend on $N$ as well as basal shear stress $\tau_b$; might this not then ruin the conclusion? The answer is no, at least for sliding laws of power law type.

(ii) Second, Schoof's 2010 paper indicates a minimum value of $N \approx 2.6$ MPa. This seems very high, and particularly seems unable to explain the very low values of $N$ seen in the Siple Coast ice streams, for example. One might say these are sediment-floored, so that the concept of bed roughness is less clear to understand: does this mean one must abandon this theory in that context? The reason I enquire is that it seems to me that the understanding of sub-ice sheet floods is something this theory should aspire to.

(iii) The boundary condition $N = 0$ is applied at the glacier snout. This is problematic because the closure equation then predicts $S$ increases indefinitely. In the Schoof 2014 paper this is circumvented by saturating the opening term as $u_b h_r \left( 1 - \dfrac{S}{S_0} \right)$, allowing for drowning of roughness at conduit size $S_0$; this allows a steady state to be reached, but one in which $S > S_0$, which makes no physical sense.

In fact, the issue with the boundary condition is that the outlet flow must become open to the atmosphere at some unknown point upstream of the snout, where I think two boundary conditions should be prescribed for $N$ and $N_x$, corresponding to continuity of water pressure and water flux. This may be important in view of figures 8 and 9, for example.

Now we come to the main issue with this paper, which lies in its style. The paper does not know whether it is for glaciologists or applied mathematicians. The message is in fact fairly simple: here is a modification of Nye which allows limit cycles, even in a lumped version, and allows leakage between floods. But the material is drawn out by over-elaborate interpretations and explanations, and veers off into dynamical systems language which is neither helpful or informative. Starting on page 6, there is a rather long-winded stability analysis, which descends by page 8 to undergraduate mathematics. The only explanation can be that this is meant for glaciologists; but my view is that if they want to learn this material they should do so in textbooks, not in a research paper. And in fact, all you need is figure 4.

It goes on: we get undergraduate discussion of Hopf bifurcation, which by page 14 has slowed to the point of somnolence. And on. The section on asymptotic solutions on page 17 is mostly out of place here. What I actually think should happen is that the paper should be rewritten in two versions: a longer mathematical one which goes to a more mathematical journal (but then suitably prunes the more elementary stuff),

and a shorter glaciological version which punches out the results: which are the model and some of the figures really.

What is missing from this is a focus on a particular flood: Summit Lake for example. Admittedly Grímsvötn has been extensively studied, and I don't know whether there are other such locations, but the lack of the possibility of even qualitative comparison to real data is a pity.

One of the issues which may deserve better attention is the occurrence of negative effective pressures, as for example in figure 8. There is some discussion of this (e. g., page 16) but it is not very satisfactory to my mind. Very high input rates can cause fracture-like floods, as in the Gjálp eruption of 1996 leading to the over-pressured jökulhlaup from Grímsvötn, but I don't know that they are that common. And according to Post and Mayo's USGS report, Summit Lake floods have been initiated below flotation. So to me this is a drawback of the model.

---

## Referee Comment (RC2) · Mauro Werder (Referee) · 17 Sep 2019

This paper investigates the mathematical properties of a conduit model (R-channel + one linked-cavity) when the upper boundary condition is a reservoir of fixed surface area and recharge rate. It looks at reservoir sizes corresponding a range from large glacier dammed lakes to moulins. It finds that for a given reservoir size that there are two stable regimes: one when the reservoir drains through the linked-cavity (i.e. recharge is low) and the other, when recharge is high, it drains through a moulin-like configuration. In-between the reservoir drainage is unstable and in fact periodic (for most situations), i.e. lake outburst floods.

I think the paper nicely illustrates and investigates the range of behaviours to be ex-
pected from ice dammed reservoirs. Whilst this range of behaviours (leaking lake, out-bursting lake, and moulin-like) has been known, it has not yet been described quantitatively; certainly not with this mathematical rigour. Thus this paper is a significant step forward.

However, the paper is a bit on the technical side for a glaciological paper. This is nicely illustrated by the brutal subsection of the Discussion (Sec 5.2), where the un-expecting reader – suffering from formula-overload already – is again presented with a wall of maths. And all this "discussion" does not actually lead to any further notable points of Discussion.

I recommend to publish this MS after Sec 5.2 has been banned to the supplement (or another publication) and the comments detailed below are addressed.

**Scientific comments**

The different triggering mechanism should probably be discussed a bit further. Of interest is in particular that many lakes drain with different triggering mechanisms from outburst to outburst (e.g. Grimsvötn 1996 vs other years, Gorner Lake (Huss et al., 2007)). In the case of Gorner Lake, no observations can predict the triggering mechanism.

Further, Huss et al. (2007) also show that Gorner Lake can indeed leak before drainage. This should be mentioned alongside Fisher (1973).

The paper by Kessler and Anderson (2004) should be discussed further, both in the Introduction and Discussion as it uses also a conduit model (linked-cavity + R-channel) and applies it to a lake drainage (their section 4.2). For instance, they also see the pre-drainage leakage.

The model used has a single cavity, but it could also be used with many cavities in parallel. The supp. of Schoof (2010) does this. Mention briefly what the impact would be, I suspect it would only be quantitative.

The fact that moulins are "small reservoirs" is only mentioned really late in the MS. Could/should this be mentioned earlier?

It is not clear to me why Appendix A is there but most other extra calculations are in the supplement, in particular as the more detailed calculation of Appendix A is also in the supplement.

**Specific comments**

Please run a spell checker over the MS!

P1, L7: delete "a"

P1, L16: mention that a lake drainage can also terminate when the lake is completely empty. This should be mentioned at a later stage as well, stating that this is not not relevant for this MS (the bed is flat).

P1, L20 write: "magnitude and timing of the flood." As for hazard prevention knowing the timing is probably equally important to the magnitude.

P2, L13: "directly directed" is awkward

Eq 1a: state that the pressure dependence of the melting point is neglected

Eq 1c: I find this equation strange. For $v_o$ (and $v_c$) no separate equation $v_o = v_o(S)$ is added either. Thus be consistent and just write $q(S, \Psi)$ in Eq 1a&b. Similarly in all later equations.

P2: would it make sense to somewhere define what a "conduit", a "channel" and a "cavity" is? For example on P7,L4 "conduit" is used signalling the use of the $v_o$ term again. So an unexpected reader may trip there without a clear definition.

P4, L5: write "background hydraulic potential"

P4, L24: "large lakes"

P4, L28: This paragraph confuses me. Is this not obvious? If not, be explicit what is

odd. If it is obvious, delete.

P5, L1: "in general"

Figure 0: A figure depicting the used geometry would be helpful.

P6, L28: "model"

P7, L21: these "reservoirs" are, e.g., moulins. Why not state this here?

P8 (on this page line numbers are messed up), L5+2: $q(\bar{S}, \bar{N})$

P8, first un-numbered Eq: this should be $v_o$ not $v_0$. Or is $v_0 = v_o(S)$? If so state.

P9, L24-29: this describes again a moulin

Fig 1: replace "lake" with "reservoir"

Fig 1: parenthesis missing after "3.3"

P11, L11: again $v_o$

P12, L22–23: the "immediately" needs to be weakened here. According to fig. 3: lake is empty and starts filling at the point (70,10), then the lake is filling again but $S$ still drops from 10m$^2$ to $\sim$0.

Fig.3: Split the second sentence at the "and"

Fig.4: could the plotted $V_p$ be added as horizontal lines to Fig. 6?

Fig.4: I don't understand what the line style "solid dashed" is supposed to be. I think the unstable periodic should be described as "dotted coloured"

Fig.4: "insets"

Fig.5: zoom to relevant $q_{in}$ values

Fig.6: it is not clear to me what is meant with "as well as in a small strip to the right of the right-hand branch of the red curve." nor is this intriguing strip ever mentioned in the

text. Clarify. Maybe a zoomed inset?

Eq 16: should be "$\sim$"

P19, L16: "the its", delete "the"

Fig 7: "(b), 10 km"

Fig 7: "plotted in black" needs to be more specific. "plotted as black line"

Fig. 89, P32, L5: In view of this model behaviour and the potentially unstable nu-merics, some words should be said about the employed numerical methods (spatial discretisation, time-stepper). Yes, the code is provided but this should be in the text.

Fig 9: What is $S_R$?

P23, L8: I don't see: "amplitude slowly grows" in fig 10

P25, L1: "reservoir" instead of "lake"

P26: twice wrong reference to 2012 instead of 2014

P26, L6: "results"

P26, L21: "a spatial"

P27, L11: $\mathcal{R}$ needs to be defined

P30, L5–9: remove if Sec 5.2 is removed

Supplement: Excellent, that the code is published! Two things: (1) There should be a README in each zip file, stating at least which script needs to be run to produce which figures. (2) I would suggest to add a licence to each zip-file (preferably an approved open source licence, the BSD-licence is popular with Matlab files https://opensource.org/licenses/BSD-3-Clause). Then it is clear under which conditions the code can be used.

---

## Author Comment (AC2) · 12 Nov 2019

Christian Schoof

cschoof@eos.ubc.ca

*This paper investigates the mathematical properties of a conduit model (R-channel + one linked-cavity) when the upper boundary condition is a reservoir of fixed surface area and recharge rate. It looks at reservoir sizes corresponding a range from large glacier dammed lakes to moulins. It finds that for a given reservoir size that there are two stable regimes: one when the reservoir drains through the linked-cavity (i.e. recharge is low) and the other, when recharge is high, it drains through a moulin-like configuration. In-between the reservoir drainage is unstable and in fact periodic (for most situations), i.e. lake outburst floods. I think the paper nicely illustrates and investigates the range of behaviours to be expected from ice dammed reservoirs. Whilst this range of behaviours (leaking lake, out-bursting lake, and moulin-like) has been known,*

[Figure]

*it has not yet been described quantitatively; certainly not with this mathematical rigour. Thus this paper is a significant step forward. However, the paper is a bit on the technical side for a glaciological paper. This is nicely illustrated by the brutal subsection of the Discussion (Sec 5.2), where the un-expecting reader – suffering from formula-overload already – is again presented with a wall of maths. And all this "discussion" does not actually lead to any further notable points of Discussion. I recommend to publish this MS after Sec 5.2 has been banned to the supplement (or another publication) and the comments detailed below are addressed.*

I would like to retain some version of the material in sec 5.2 here as opposed to offloading it to another paper, since such a paper would be rather stunted. The purpose of 5.2 was two-fold: first, the stability analysis leans heavily on that already presented in section 3.1, so keeping it in the same paper (overall, be that in the main text or supplementary material) makes sense to me. Secondly, the earlier paper by Schoof et al (2014) provides the main motivation for the present paper, and there we discuss not only outburst floods from localized reservoirs but also the effect of spatially extended storage, so it is natural to tie together the analysis of the mechanisms invovled in causing unsteady flow. I would propose a solution where most of section 5.2 goes into the supplementary material but I still sketch the main insights (mostly in words) in the discussion section, tying this better into the rest of the discussion.

*The different triggering mechanism should probably be discussed a bit further. Of interest is in particular that many lakes drain with different triggering mechanisms from outburst to outburst (e.g. Grimsvötn 1996 vs other years, Gorner Lake (Huss et al., 2007)). In the case of Gorner Lake, no observations can predict the triggering mechanism.*

This I am happy to do.

*Further, Huss et al. (2007) also show that Gorner Lake can indeed leak before drainage. This should be mentioned alongside Fisher (1973).*

Ditto

*The paper by Kessler and Anderson (2004) should be discussed further, both in the Introduction and Discussion as it uses also a conduit model (linked-cavity + R-channel) and applies it to a lake drainage (their section 4.2). For instance, they also see the pre-drainage leakage.*

Ditto

*The model used has a single cavity, but it could also be used with many cavities in parallel. The supp. of Schoof (2010) does this. Mention briefly what the impact would be, I suspect it would only be quantitative.*

Again, I am happy to do that. Things get a touch more complicated here and ultimately reveal the limitation of a 'lumped' model in the sense that a spatially extended (or 2D network model) predicts the expected formation of a single channel, whereas multiple identical conduits in parallel as in Schoof (2010) rather strangely predict that the stable limit cycle involves all conduits eventually behaving identically: the channelizing instability that causes one conduit to 'win' apparently does not work fast enough in the simplified, lumped-conduits-in-parallel world the comment refers to. Exactly why I have not figured out yet, but I can do a little further work to elaborate on this - most likely also to be banished to the supplementary material.

*The fact that moulins are "small reservoirs" is only mentioned really late in the MS. Could/should this be mentioned earlier?*

Sure

*It is not clear to me why Appendix A is there but most other extra calculations are in the supplement, in particular as the more detailed calculation of Appendix A is also in the supplement.*

Appendix A is there to make the paper a little more self-contained. I felt (and this perhaps aligns with referee # 1?) that the other material in the supplementary material is

really designed to fill in detail that a reader familiar with basic theory of dynamical systems could reconstruct, should they really want to try (in the terms of the other review, this is material that I'm including to help out the more "glaciological" reader, which a "mathematical" reader would probably not need). Appendix A is a brief summary of the material that goes beyond that level - it provides a brief sketch of results in the main paper that I expect would not be totally straightforward to reconstruct for a "mathematical" reader without undue effort.

*Please run a spell checker over the MS!*

Yes. This seems to be a common complaint, my apologies!

*P1, L7: delete "a"*

sure

*P1, L16: mention that a lake drainage can also terminate when the lake is completely empty. This should be mentioned at a later stage as well, stating that this is not not relevant for this MS (the bed is flat).*

Indeed; I was focusing on the "other", lower bound on $N$ (zero) because it plays a role in initiation, instead of on the uppper (lake empty) bound because it aligns more closely with the main thrust of the paper, but fair point.

*P1, L20 write: "magnitude and timing of the flood." As for hazard prevention knowing the timing is probably equally important to the magnitude.*

Ok

*P2, L13: "directly directed" is awkward*

will omit "directly"

*Eq 1a: state that the pressure dependence of the melting point is neglected*

sure

*Eq 1c: I find this equation strange. For vo (and vc ) no separate equation vo = vo (S) is added either. Thus be consistent and just write q(S, Ψ) in Eq 1a & b. Similarly in all later equations.*

The reason for doing this is in the form stated is that $q$ is a variable, not a function (ie there is a $\partial q / \partial x$, which doesn't make sense if $q$ is a function (not of $x$) — the inconsistency is rather in using the same letter for variable and function. I will change this.

*P2: would it make sense to somewhere define what a "conduit", a "channel" and a "cavity" is? For example on P7,L4 "conduit" is used signalling the use of the vo term again. So an unexpected reader may trip there without a clear definition.*

Sure, I will elaborate on this to the effect that a conduit is a generic drainage element, while channels and cavities are really equilibria of a conduit that have different qualitative properties. My mistake was probably being careless about the use of the word "channel" (e.g. p. 4, line 11) — I will now flag at the start of section 2 that I will use "conduit" consistently (using channel-like and cavity-like to describe situations in which there are dominant balances that are like those in classical channel and cavity models), and make sure that I actually follow that terminology consistently.

*P4, L5: write "background hydraulic potential"*

sure

*P4, L24: "large lakes"*

yes

*P4, L28: This paragraph confuses me. Is this not obvious? If not, be explicit what is odd. If it is obvious, delete.*

Nothing is odd here, but "obvious" is probably in the eye of the beholder. I put this in for the reader not as familiar with the theory of outburst floods

*P5, L1: "in general"*

indeed

*Figure 0: A figure depicting the used geometry would be helpful.*

Ok

*P6, L28: "model"*

yes

*P7, L21: these "reservoirs" are, e.g., moulins. Why not state this here?*

will do; there is of course the case of very large reservoirs too, which are probably not realistic but theoretically can be stabilized, so I don't want to make a one-to-one correspondence with moulins here.

*P8 (on this page line numbers are messed up), L5+2: q(SÌĎ, NÌĎ )*

indeed

*P8, first un-numbered Eq: this should be $v_o$ not $v_0$ . Or is $v_0 = v_o(S)$? If so state.*

This is a simple typo.

*P9, L24-29: this describes again a moulin*

I am happy to mention moulins here, subject to the same caveat as above

*Fig 1: replace "lake" with "reservoir"*

ok

*Fig 1: parenthesis missing after "3.3"*

indeed

*P11, L11: again $v_o$*

correct

*P12, L22–23: the "immediately" needs to be weakened here. According to fig. 3: lake is empty and starts filling at the point (70,10), then the lake is filling again but S still drops from 10m2 to âĹij0.*

This is true, but the it takes a very short amount of *time* (rather than distance in phase space) for that gap in phase space to be covered. I will say 'almost immediately'

*Fig.3: Split the second sentence at the "and"*

Good idea

*Fig.4: could the plotted Vp be added as horizontal lines to Fig. 6?*

No doubt they can.

*Fig.4: I don't understand what the line style "solid dashed" is supposed to be. I think the unstable periodic should be described as "dotted coloured"*

Indeed. I have no idea what I was thinking. "Dotted coloured" it is.

*Fig.4: "insets"*

yes

*Fig.5: zoom to relevant $q_{in}$ values*

Yes, I will get rid of the blank space on the left

*Fig.6: it is not clear to me what is meant with "as well as in a small strip to the right of the right-hand branch of the red curve." nor is this intriguing strip ever mentioned in the text. Clarify. Maybe a zoomed inset?*

I will clarify. This is the small range of $q_{in}$ values in figure 4 where a limit cycle co-exists with a stable equilibrium (and an unstable periodic solution), see e.g. the inset in figure 4b.

*Eq 16: should be "âĹij"*

yes

*P19, L16: "the its", delete "the"*

yes

*Fig 7: "(b), 10 km"*

yes

*Fig 7: "plotted in black" needs to be more specific. "plotted as black line"*

indeed, will change

*Fig. 8/9, P32, L5: In view of this model behaviour and the potentially unstable numerics, some words should be said about the employed numerical methods (spatial discretisation, time-stepper). Yes, the code is provided but this should be in the text.*

I will make clear that the numerics are described in Schoof et al (2014, appendix)

*Fig 9: What is $S_{\mathcal{R}}$ ?*

It's an idiotic legacy notation error on my part; it should say $S$.

*P23, L8: I don't see: "amplitude slowly grows" in fig 10*

Ok, "slowly" may be an overstatement. "Grows and saturates" may be better

*P25, L1: "reservoir" instead of "lake"*

thanks for spotting this

*P26: twice wrong reference to 2012 instead of 2014*

yes

*P26, L6: "results", P26, L21: "a spatial"*

indeed

*P27, L11: $\Re$ needs to be defined*

Ok; would plain "Re" be ok as "real part"

*P30, L5–9: remove if Sec 5.2 is removed*

see above

*Supplement: Excellent, that the code is published! Two things: (1) There should be a README in each zip file, stating at least which script needs to be run to produce which figures. (2) I would suggest to add a licence to each zip-file (preferably an approved open source licence, the BSD-licence is popular with Matlab files https://opensource.org/licenses/BSD-3-Clause). Then it is clear under which conditions the code can be used.*

Will do

---

## Author Response (AR1)

**An analysis of instabilities and limit cycles in glacier-dammed reservoirs: response to reviewers**

**Christian Schoof**

**1 Referee # 1**

This is an awkward paper to referee. Being Christian Schoof, there is not going to be anything technically wrong with it, but from my perspective it is not properly thought out.

Nor is it well presented. The very first thing to say is that the text is littered with typographical and grammatical errors, so
many that I will not list them all here. But there are such errors on page 1, line 16; 1,24; 2,5; 2,6; 2,13; 2,15; 2,20 (twice); 2,29 (the whole second half); 2,35; and so on, and on, and on. Perhaps the best is saved for last, where Schoof's own 2012 paper is mis-referenced (it is part 1, not part 2). Incidentally, all the poor authors cited in the references are demoted to a single initial each.

I'm not sure my mere name warrants such confidence in the results, but I'll take this to mean I actually haven't made any

- 10 mistakes. The apparent awkwardness is as difficult to answer at this point as it may have been for the referee to review the paper in the first place. In view of this, I will respond to the specific comments later. As far as the presentation is concerned or rather, the number of typos and other superficial but annoying errors — I'm happy to concede the paper may have been submitted in too much of a hurry. My apologies for that. With regard to the author initials, I'll simply point out that I am using the Copernicus bibliography style file. As such the complaint is probably best addressed to the publisher.
- 15 The paper concerns a model (which is analysed in both a 'lumped' form and a spatially dependent one) for subglacial floods, or jokulhlaups. The paper is motivated heavily by previous work of Fowler and Ng, and seeks to modify this earlier work, by allowing for the case where there is no 'seal' of the subglacial (or ice-dammed) lake, which can then continually leak between floods, as is the case for Summit Lake, according to Fisher in 1973. Note: Fisher, not Fischer.

Fair point. I must have allowed my teutonic roots to influence my spelling.

20 The improvement consists of showing that with an extra term in the closure equa- tion of the classical Nye-Rothlisberger theory, the model will describe limit cycles even in the absence of a seal. This seems to me the principal achievement of this paper. The extra term invoked is an ingenious addition due to Schoof in 2010 which allows the description of both cavity drainage and channel drainage within the confines of a single model. It is worth offering some comments on this addition.

Indeed. I'd hope that beyond the qualitative statement that limit cycles are possible by adding a mechanism by which the drainage system remains 'open' in the refilling phase, an analysis of how flood magnitude and timing depends on forcing and geometry a valuable, too. In its original form, the extra term appears as the first term on the right hand side of the closure equation

5

10

$$\dot{S} = u_b h_r + c_1 \Psi Q - c_2 N^n \left(\frac{Q}{c_3 \sqrt{\Psi}}\right)^{0.8}.$$

and it describes the opening of cavities by ice flow (velocity  $u_b$ ) over bedrock bumps (height  $h_r$ ). The steady state of this equation provides for both channels (N increases with Q) and linked cavities (N decreases as Q increases). There are several comments to make.

(i) We might suppose in reality that ub will itself depend on N as well as basal shear stress  $\tau_b$ ; might this not then ruin the conclusion? The answer is no, at least for sliding laws of power law type.

This is true; I have deliberately skirted this issue as the kind of glacier junctions at which dams are likely to occur often have awkward geometries — this is certainly true for Summit Lake, or the field site at the Kaskawulsh Glacier that has motivated my own interest in this subject (and before you ask: I do intend to publish data from said site, but inclusion in this paper would sure; y break the bounds of what is reasonable for paper length, even if I were to shorten the analysis). Those awkward geometries matter in the sense that treating ice flow as a function of local N might be stretching credulity. As the reviewer rightly points out, making  $u_b$  dependent on N doesn't ultimately break the mechanism being investigated, so long as  $u_b$  remains bounded below by something greater than zero. It just makes the flood cycle even less simple to describe. I'd be happy to add a brief discussion to the supplementary material if desired, but don't think this will add much to the main paper.

(ii) Second, Schoof's 2010 paper indicates a minimum value of N 2.6 MPa. This seems very high, and particularly seems
unable to explain the very low values of N seen in the Siple Coast ice streams, for example. One might say these are sediment-floored, so that the concept of bed roughness is less clear to understand: does this mean one must abandon this theory in that context? The reason I enquire is that it seems to me that the understanding of sub-ice sheet floods is something this theory should aspire to.

This, I believe, is actually a reference to the fact that classical R-channel models with Nye (1953)-type closure rates consistently overpredict effective pressures, not just in the sense that they would require effective pressures larger than overburden, but in the sense that *measured* overburden pressures are usually significantly smaller. That is not an original observation of mine. The main reference to this that I'm aware of is Hooke, R.LeB. and Laumann, T. and Kohler, J., 1990, Subglacial water pressures and the shape of subglacial conduits. *J. Glaciol.* 36(122), 67–71. The point is that flatter channel shapes allow smaller effective pressures to balance predicted dissipation rates in the flow, and therefore to reconcile observation and theory. In the context of the model being used here, this issue is discussed at length in Schoof (2014) as cited in the present manuscript.

I agree that this is an appealing direction to develop outburst flood models in, and in particular, that the question of what the lateral aspect ratio of a channel actually is deserved further attention (ideally building on the vastly underappreciated D.Phil. thesis by Felix Ng.) Probably beyond the scope of this paper, though. As far as the concept of bed roughness becoming nebulous for deformable beds is concerned, I'm inclined to agree for relatively fine-grained beds with a narrower grain size

30 distributions — as would apply for the formerly submarine bed areas of West Antarctica, for instance. For polydisperse grain size distributions, where there are larger cobbles and boulders mixed into the till, I'd argue that there are likely to be bed protrusions that can support cavities as in the canonical hard bed picture.

(iii) The boundary condition N = 0 is applied at the glacier snout. This is problematic because the closure equation then predicts S increases indefinitely. In the Schoof (2014) paper this is circumvented by saturating the opening term as  $u_b h_r (1 - S/S_0)$ , allowing for drowning of roughness at conduit size S0; this allows a steady state to be reached, but one in which S > S0, which makes no physical sense. In fact, the issue with the boundary condition is that the outlet flow must become open to the

5 atmosphere at some unknown point upstream of the snout, where I think two boundary conditions should be prescribed for N and  $N_x$ , corresponding to continuity of water pressure and water flux. This may be important in view of figures 8 and 9, for example.

What is really at issue here I think is the way the cavity opening rate "goes away" for large S (note that this is only relevant to the spatially extended model, so I will restrict my discussion to the latter). In the model as posed, as in the earlier Hewitt et

- 10 al (2012), Schoof (2010) and Schoof et al (2012, 2914) papers, this is done by writing the opening rate as  $u_b(h_r h)/l_r$ ) for a continuum "sheet", or equivalently as  $u_bh_r(1 - S/S_0)$  for an individual conduit. This does have the somewhat unintended consequence of leading to the opening rate becoming negative in an unbounded way when S exceeds the threshold  $S_0$ . A better way of dealing with the idea that bed roughness cannot indefinitely lead to a constant opening rate as conduit size grows (which is probably robust) might be to write the opening rate as  $u_bh_rf(S/S_0)$  with  $f(x) \to 1$  as  $x \to 0$  and  $f(x) \to 0$  as
- 15  $x \to \infty$ ; something like  $f(x) = (1 + \tanh(x))/2$  would do.

Having made the choice of cut-off function we have made here (where f goes to  $\infty$  linearly as  $x \to 0$  instead of vanishing), we can ask what difference this makes. In a model where cavity opening vanishes for large conduit sizes, the dominant balance near a glacier margin where  $N \to 0$  would be between the melt rate  $c_1 Q \Psi$  and closure rate  $c_2 S N^n$ . This would still leave the problem singular at the margin with  $S \to 0$ , but not in a pathological way (and the problem could further be regularized to maintain finite S by supposing that the glacier ends in a cliff so N is small but finite, or by supposing that the channel evolution equation is not cast in terms of  $\partial S/\partial t$  but the material derivative  $\partial S/\partial t + u_b \partial S/\partial x$ .1

20

For that case, we can construct a near-margin form of the solution, and further ask how different the solution to other versions of the model is. In particular, for a pure channel model with a vanishing cavity opening rate, we get

$$S_t = c_1 Q_{\Psi} - c_2 S |N|^{n-1} N, \qquad Q = c_2 S^{\alpha} |\Psi|^{-1/2} \Psi, \qquad \Psi = \Psi_0 + N_x$$

and given a fixed discharge Q, a steady-state near-terminus solution can be constructed by noting that

$$\Psi = \left(Q/(c_3 S^\alpha)\right)^{1/2}$$

so

$$c_1 c_3^{-2} Q^3 S^{-(2\alpha+1)} = c_2 N^n$$

and hence

$$N_x = -\Psi_0 + c_1^{-2\alpha/(2\alpha+1)} c_3^{-2/(2\alpha+1)} Q^{(1-4\alpha)/(2\alpha+1)} (c_2 N^n)^{2\alpha/(2\alpha+1)}$$

<sup>1This idea is due to Ian Hewitt, who may indeed have published it somewhere. While unappealing for cavities that are tied to bed roughness, the advection term must play a role near the snout, where melting happens not only because of subglacial water flow, but also from the surface, and ice flow must compensate for that.

which is clearly solvable form x < L with a boundary condition N(L) = 0; the near field behaves as

$$\begin{split} N \sim & \Psi_0(L-x) - \\ & \frac{2\alpha+1}{2\alpha n+2\alpha+1} c_1^{-2\alpha/(2\alpha+1)} c_2^{2\alpha/(2\alpha+1)} c_3^{-2/(2\alpha+1)} Q^{(1-4\alpha)/(2\alpha+1)} \Psi_0^{2\alpha n/(2\alpha+1)} (L-x)^{(2\alpha n+2\alpha+1)/(2\alpha+1)} \\ & S \sim & c_1^{1/(2\alpha+1)} c_2^{-1/(2\alpha+1)} c_3^{-2/(2\alpha+1)} Q^{3/(2\alpha+1)} \Psi_0^{-n/(2\alpha+1)} (L-x)^{-n(2\alpha+1)} \end{split}$$

Clearly, we can see that N remains well-behaved (and indeed positive, so the channel need not be partially open to the atmosphere!) while S blows up in a power-law fashion. We can go further and ask what the stability properties of the channel-only problem look like in the near field and construct a linearization. This is best done by changing the dependent variable S into something that remains bounded. An obvious choice is

$$Y = S^{1-\alpha}$$

5 The dominant balance when linearizing the problem above about the steady state  $(Y = \bar{Y} + Y', N = \bar{N} + N', \Psi = \bar{\Psi} + N'_x, Q = \bar{Q} + Q'$ , where barred quantities are steady state solutions and primed quantities are small perturbations) works out to be  $Y'_t \sim \frac{3}{2}c_1c_3\bar{\Psi}^{1/2}N'_x$  $Q' \sim \frac{\bar{Q}}{2\bar{\Psi}}N'_x$

The germane question with using different model formulations that do not suppress the cavity opening term as above is
whether they lead to the same solution away from a small region near the margin. As in, does the "regularization" of the model make any difference? In view of the question about numerical results tin figures 8 and 9, the question I will try to address is whether discrepancies between the channel-only model advocated above and the model used in the paper become more pronounced at large water throughputs in the model, which is the parameter regime that these calculations look at. For more moderate throughputs, the good agreement between lumped and spatially extended model suggests the issue of what happens near the margin (which does not feature in the lumped model) becomes less relevant.

The model used in the paper replaces the above by

$$S_t = c_1 Q_{\Psi} + u_b h_r (1 - S/S_0) - c_2 S|N|^{n-1}N, \qquad Q = c_2 S^{\alpha} |\Psi|^{-1/2} \Psi, \qquad \Psi = \Psi_0 + N_x$$

In order to look at the difference from the channel-only model obtained by putting  $u_b h_r = 0$ , I will scale this by defining

$$[S] = \left(\frac{[Q]}{c_3[\Psi]^{1/2}}\right)^{1/\alpha}, \qquad [t] = \frac{[S]}{c_1[Q][\Psi]}, \qquad [N] = \left(\frac{c_1[Q][\Psi]}{c_2[S]}\right)^{1/n}, \qquad [x] = \frac{[N]}{[[\Psi]]}$$

where [Q] and  $[\Psi] = \Psi_0$  are assumed to be given. Putting

$$S^* = \frac{S}{[S]}, \qquad N^* = \frac{N}{[N]}, \qquad \Psi^* = \frac{\Psi}{[\Psi]}, \qquad Q = \frac{Q}{[Q]}, \qquad t^* = \frac{t}{[t]}, \qquad x^* = \frac{x}{[x]},$$

and immediately dropping the star decorations, the model becomes

 $S_t = Q\Psi + \delta - \nu S - S|N|^{n-1}N, \qquad Q + S^{\alpha}|\Psi|^{-1/2}\Psi, \qquad \Psi = 1 + N_x$

where

$$\delta = \frac{u_b h_r}{c_1[Q][\Psi]}, \qquad \nu = \delta[Q]^{1/\alpha} c_3^{-1/\alpha} [\Psi]^{-1/(2\alpha)} S_0^{-1}.$$

We can repeat the exercise of finding steady states. Assuming without loss of generality that the scaled flux Q = 1, we find  $\Psi = S^{-2\alpha}$  and

$$S^{-(2\alpha+1)} + \delta/S - (nu+N^n).$$

Hence

$$S = (nu + N^n - \delta/S)^{-1/(2\alpha+1)},$$

$$N_x = -1 + (\nu + N^n - \delta/S)^{-1/(2\alpha+1)}$$

which the channel only model replaces by

5
$$S = (nu + N^n)^{-1/(2\alpha+1)},$$

 $N_x = -1 + (\nu + N^n)^{-1/(2\alpha+1)}$

We want to know whether for larger |L - x|, the full and channel-only models will agree. This will be the case provided N agrees between the two models, and the latter will be the case if the correction  $\delta/S$  remains small compared with  $nu + N^n$  as well as having  $\nu \ll 1$ . This will be the case so long as  $S \sim \nu^{-1/(2\alpha+1)}$  near x = L is large enough, in other words, if

10 1 ≫ ν ≫ δ/nu-1/(2α+1) or ν ≪ δ2α/(2α+1). The definitions of δ and ν above show that ν/δ-2α/(2α+1) increases with [Q], all other parameters being constant, so we would in fact expect closer agreement between full and channel-only models for large [Q].

We can go further and look at the linearization of the problem, again in terms of the variable Y used above (or rather, its obvious dimensionless counterpart); the dominant balances when adding the cavity opening term become

$$\begin{array}{ll} {\rm 15} & Y_t' \sim & \frac{3}{2} \bar{\Psi}^{1/2} N_x' + \delta \bar{Y}^{1/(\alpha-1)} Y' \\ & Q' \sim & \frac{1}{2 \bar{\Psi}} N_x' \end{array}$$

By similar construction to the above, if the steady state converges to that for the channel-only model as |L - x| becomes large, so will the linearized solution for small  $\delta$ ; in other words, the additional term due to cavity opening will remain a small correction. This suggests that the stability results in the main paper remain robust for large water throughput rates.

20

- Now we come to the main issue with this paper, which lies in its style. The paper does not know whether it is for glaciologists or applied mathematicians. The message is in fact fairly simple: here is a modification of Nye which allows limit cycles, even in a lumped version, and allows leakage between floods. But the material is drawn out by over-elaborate interpretations and explanations, and veers off into dynamical systems language which is neither helpful or informative. Starting on page 6, there is a rather long-winded stability analysis, which descends by page 8 to undergraduate mathematics. The only explanation can
- 25 be that this is meant for glaciologists; but my view is that if they want to learn this material they should do so in textbooks, not

in a research paper. And in fact, all you need is figure 4.

It goes on: we get undergraduate discussion of Hopf bifurcation, which by page 14 has slowed to the point of somnolence. And on. The section on asymptotic solutions on page 17 is mostly out of place here. What I actually think should happen is that the paper should be rewritten in two versions: a longer mathematical one which goes to a more mathematical journal (but then

5 suitably prunes the more elementary stuff) and a shorter glaciological version which punches out the results: which are the model and some of the figures really.

Style may be where the referee and I won't agree. I am happy to shorten some of the material in the paper where appropriate, such as the linear stability analysis. The existing text undoubtedly can be optimized in that sense, but I don't think that's the issue. I understand the rationale for splitting work between "mathematical" papers and "glaciology" papers. This has been

- 10 practised by a number of researchers in the past (Hutter, Morland, Fowler etc.) and even I have been known to try. However, in my own experience, what happens is that these mathematical papers, to the extent that they are taken up by anyone, get cited by glacioloists, not by applied mathematicians or fluid dynamicists outside of glaciology. The only exception are perhaps those dealing with numerical analysis of glaciological partial differential equations. A brief trawl through an indexing website like Web of Science should confirm that impression.
- 15 In short, there seems little point in these separate mathematical papers for an imaginary specialist audience. At the same time, I do not believe in simply saying "here are our mathematical results, but you wouldn't really understand so we won't explain any of the detail", which is the risk I see in writing a "glaciological version". What I do see in glaciology is an increasing number of researchers who have solid background in physics or similar disciplines. These researchers have the ability to understand mathematical material but may need a more didactic approach than the simple assumption that they have
- 20 not only taken a course in dynamical systems theory, but actually remember its contents. This is the audience I'd like to reach here. Yes, doing so may mean a more pedestrian pace for the fully-fledged mathematician as a reader, but there are few enough of those around that I'm disinclined to worry (except about the referee, who I assume is an applied mathematician). I should add: I understand that a paper is not a textbook, but slightly more explanation to get a point across does not go amiss, and I think the manuscript as submitted is honest about what is ultimately textbook material and what is not (although Stogatz may
- admittedly be a more suitable textbook for the target audience than Wiggins).

I would add that a 'didactic approach' to presenting mathematical material in glaciology has been taken previously, even where that material arguably has limited novelty in a global (as opposed to discipline-specific) sense: to name but one example, a number of papers published in the Journal of Glaciology around 2011 (primarily by Bassis and Dukowicz et al) have elaborated on the fact that Stokes' equations are equivalent to a minimization problem — something that had been known to applied

30 mathematics and fluid dynamics at least since the 1960s, but was apparently not widely known in glaciology. Whether the referee (who presumably hails form an applied mathematics background) would regard those papers as giving an undergraduate introduction to the calculus of variations I can't tell, but these particular papers clearly have had some impact (with 8 and 30 citations, respectively).

**An analysis of instabilities and limit cycles in glacier-dammed reservoirs: response to reviewers**

**Christian Schoof**

**1 Referee # 2**

25

**Referee comment:** This paper investigates the mathematical properties of a conduit model (*R*-channel + one linked-cavity) when the upper boundary condition is a reservoir of fixed surface area and recharge rate. It looks at reservoir sizes corresponding a range from large glacier dammed lakes to moulins. It finds that for a given reservoir size that there are two stable

- 5 regimes: one when the reservoir drains through the linked-cavity (i.e. recharge is low) and the other, when recharge is high, it drains through a moulin-like configuration. In-between the reservoir drainage is unstable and in fact periodic (for most situations), i.e. lake outburst floods. I think the paper nicely illustrates and investigates the range of behaviours to be expected from ice dammed reservoirs. Whilst this range of behaviours (leaking lake, out-bursting lake, and moulin-like) has been known, it has not yet been described quantitatively; certainly not with this mathematical rigour. Thus this paper is a significant step
- 10 forward. However, the paper is a bit on the technical side for a glaciological paper. This is nicely illustrated by the brutal subsection of the Discussion (Sec 5.2), where the un-expecting reader suffering from formula-overload already is again presented with a wall of maths. And all this "discussion" does not actually lead to any further notable points of Discussion. I recommend to publish this MS after Sec 5.2 has been banned to the supplement (or another publication) and the comments detailed below are addressed.
- I would like to retain some version of the material in sec 5.2 here as opposed to offloading it to another paper, since such a paper would be rather stunted. The purpose of 5.2 was two-fold: first, the stability analysis leans heavily on that already presented in section 3.1, so keeping it in the same paper (overall, be that in the main text or supplementary material) makes sense to me. Secondly, the earlier paper by Schoof et al (2014) provides the main motivation for the present paper, and there we discuss not only outburst floods from localized reservoirs but also the effect of spatially extended storage, so it is natural to
- 20 tie together the analysis of the mechanisms invovled in causing unsteady flow.

To address the referee's concerns while still touching on the effects of distributed drainage, I have now included the original section 5.2 in the supplementary material, and included a shortened version in the main paper. That shortened version simply sets up the problem with distributed drainage and the stability analysis, which allows me to clarify how the stability analysis is changed from the original version in section 2, and I simply state the main result, namely that an unstable wave can be formed even if the conduit is channel-like. Hopefully this is a little less formula-overload.

1

**Referee comment:** The different triggering mechanism should probably be discussed a bit further. Of interest is in particular that many lakes drain with different triggering mechanisms from outburst to outburst (e.g. Grimsvötn 1996 vs other years, Gorner Lake (Huss et al., 2007)). In the case of Gorner Lake, no observations can predict the triggering mechanism.

I have updated the introduction to reflect more of the observational literature on outburst floods, particularly covering 5 Grímsvötn, Hidden Creek Lake, Gornersee, and an unnamed lake at the Kaskawulsh Glacier:

One possibility for flood initiation is that the lake simply fills to the level at which the ice dam starts to float, and a sheet flow emerges between ice and bed that subsequently channelizes (Flowers et al, 2004). This may occur irregularly in same lakes such as Grímsvötn due to exceptionally large inflow rates to the lake, for instance during volcanic eruptions, or as part of a repeating flood cycle in others (Bigelow et al, 2020). Some lakes are however known to initiate outburst floods before they

10 reach that flotation level.

[...]

Grímsvötn typically starts its flood when water levels are below flotation (Björnsson, 1992), without successive floods growing in amplitude, except when the flood results from a large increase in water input due to volcanic activity (Gudmundsson et al, 1997). To explain this behaviour, Fowler (1999) considers the effect of a water supply along the length of the channel on its

15 evolution. Such a water supply can maintain a minimum channel size even between outburst floods, with flow of water being directed partly into the lake and partly down-glacier to the margin, provided the glacier has a geometrical 'seal'. As the lake fills, the flow divide inside the channel migrates towards the lake, and the flood begins when the divide reaches the edge of the lake.

While this mechanism successfully explains how limit cycles (stable, periodic oscillations in lake level) can emerge in the

- 20 model, it also predicts that no water can leave the lake between floods. Tracer experiments conducted at Salmon Glacier in Canada (Fisher, 1973) demonstrate that lakes can leak continuously throughout their recharge phase. Summit Lake, dammed by the Salmon Glacier, also has a history of flood initiation at lake levels below the flotation level (Post and Mayo, 1971). At Gornersee in Switzerland, observations of water lake levels and inferences of lake water balance based on meteorological measurements over two consecutive years also suggest leakage and flood initiation below flotation level during one year, and
- 25 flood initiation at or above flotation level during the previous year, potentially in the absence of any pre-flood leakage (Huss et al, 2007). Observations at Hidden Creek Lake (Anderson et al 2003) by contrast suggest no leakage prior to flood initiation, though that conclusion is again based on water balance estimates rather than direct tracer experiments.

**Referee comment:** The paper by Kessler and Anderson (2004) should be discussed further, both in the Introduction and Discussion as it uses also a conduit model (linked-cavity + *R*-channel) and applies it to a lake drainage (their section 4.2). For 30 instance, they also see the pre-drainage leakage.

I have discussed this in the introduction now, stating the following:

Note that many elements of the model studied here were included in the earlier study of outburst flooding by Kessler and Anderson (2004), who were able to replicate leakage from a glacier-dammed reservoir before the onset of the outburst flood, but did not attempt to reproduce recurring floods. It is unclear their model would have been able to do so since, in common

35 with the model due to Fowler (1999), the main trunk of their drainage system was set up as a pure R-channel.

I'm quite hesitant to discuss the paper further, because I am rather uncertain whether I would be able to reproduce their model without either the code (which, in 2004, would not have been archived as supplementary material or on a code repository) or an unambiguous mathematical statement of the problem that the authors are discretizing. There are a number of unorthodox elements to the model description in Kessler and Anderson that I am somewhat loath to go into here, as I'm not addressing those authors directly, but suffice it to say that the following:

5

10

15

20

25

- 1. I *think* but cannot be sure that their network topology is tree-like, with cavity-or-channel network edges only present on the branches of the tree but not the main stem, so all water ultimately has to go through the R-channel-like main trunk, which ought therefore to behave like the R-channel in Fowler (1999). In that case, it is however unclear if there is supposed to be an analogous, uninterrupted background water supply that will keep the channel open to a minimum size and cause flow divide migration as in Fowler (1999). Knowing that would be key to understanding the initiation mechanism for *repeated* floods instead of a single flood, the latter being continent on the choice of initial conditions (which I think aren't fully specified in the paper, at least I couldn't tell). I expect the authors didn't intend to run their model for multiple melt seasons, so there may not be a unique answer to the point I'm making here. That makes discussion of their results in the context of what my manuscript tries to talk about — repeated floods — a bit difficult beyond what the current introduction states.
- 2. I also think but not entirely sure that the unorthodox description of conduit cross-section evolution in their equations (3), (4) and (6) is meant to say that the *actual* rate of change of S, i.e. dS/dt, is a potentially selective sum of the right-hand sides in (3), (4) and (5), with the main trunk missing out the right-hand side of (3); as stated, it looks like dS/dt is somehow strangely multiply-defined. The text in the present manuscript assumes this interpretation, which makes their conduits the same in principle as mine.
- 3. The description of updating h according to (1a) and (1b) in their model was also not entirely clear to me; it looks to me like (1a) and (1b) together ought to correspond to a discretized parabolic problem for h (there is a derivative of  $Q_{cc}$ with respect to a downstream variable x in (1b), and I think  $Q_{cc}$  is defined in terms of a gradient of h in (5), although the notion of a derivative is discarded there in favour of a difference, suggesting a discretization in x should equally be applied in (1b). The text however then goes on to state that h is updated sequentially from the node that initially has the highest h to lowest h (their paragraph [11]). This would be appropriate for a hyperbolic rather than parabolic problem, so again I don't feel confident I could reproduce the model

I'd rather not get into discussing these points in detail in the present manuscript.

**Referee comment:** The model used has a single cavity, but it could also be used with many cavities in parallel. The supp. of Schoof (2010) does this. Mention briefly what the impact would be, I suspect it would only be quantitative. 30

Things get a touch more complicated here: multiple identical conduits in parallel in a lumped model as in the supplementary material to Schoof (2010) perhaps unexpectedly predict that the stable limit cycle involves all conduits behaving identically. The channelizing instability that causes one conduit to 'win' when a drainage system is forced not with constant net throughput

3

of water does not work fast enough when the system is instead forced with a reservoir at the upstream end. It seems like all conduits return to basically the same size during the refilling phase, and, being buffered by the reservoir, don't actually compete effectively for flux (which conduits do under fixed net flux conditions as in Schoof (2010)).

- A linear stability analysis by itself doesn't reveal that insight. More to the point, however, it seems like allowing the conduits to have different physical parameters — for instance, different bed roughness heights or tortuosities — can fix that problem (as in, cause a single channel-like conduit to drain the lake). I intend to investigate that more deeply as part of ongoing work into surge mechanisms, but feel it would not add into insight here, and if anything, confuse the reader because it is a slightly subtle point that ultimately doesn't make a big difference, unless you suppose all channels are indeed identical. (Rather strangely, for a 2D network, a single channel seems to drain the reservoir even if all the channels are identical, and I haven't fully figured out
- 10 why yet).

**Referee comment:** The fact that moulins are "small reservoirs" is only mentioned really late in the MS. Could/should this be mentioned earlier?

I've expanded on this theme in the introduction, in the following paragraph:

The original motivation for the work in Schoof et al (2014) was to demonstrate that behaviour akin to outburst floods can occur in systems with limited or even distributed water storage, manifesting itself in the form of unforced water pressure oscillations. Such limited storage could in principle result from moulins or any other vertical shaft that can fill progressively with water as water pressure rises.such as a basal crevasse. Viewed from that alternative different perspective, the present paper analyzes a drainage model that has become widely adopted in the study of subglacial hydrology (Werder et al, 2013).

We identify instabilities that spontaneously lead to self-sustained oscillations, describing the mechanisms behind them and
20 delineating the regions in parameter space where they occur. This is likely to be useful in diagnosing the behaviour of such models, regardless of their specific application to large outburst floods emanating from easily identifiable glacier 'lakes'.

**Referee comment:** It is not clear to me why Appendix A is there but most other extra calculations are in the supplement, in particular as the more detailed calculation of Appendix A is also in the supplement.

Appendix A is there to make the paper a little more self-contained. I felt (and this perhaps aligns with referee # 1?) that the other material in the supplementary material is really designed to fill in detail that a reader familiar with basic theory of dynamical systems could reconstruct, should they really want to try (in the terms of the other review, this is material that I'm including to help out the more "glaciological" reader, which a "mathematical" reader would probably not need). Appendix A is a brief summary of the material that goes beyond that level — it provides a brief sketch of results in the main paper that I expect would not be totally straightforward to reconstruct for a "mathematical" reader without undue effort.

30 **Referee comment:** *Please run a spell checker over the MS!*

Yes. This seems to be a common complaint, my apologies!

P1, L7: delete "a"

sure

35

**Referee comment** *P1*, *L16: mention that a lake drainage can also terminate when the lake is completely empty. This should be mentioned at a later stage as well, stating that this is not not relevant for this MS (the bed is flat).*

I have added the following to the end of the first paragraph of the introduction:

Alternatively, the flood can terminate because the lake has run dry. Then conduit then becomes partially air-filled and closes with minimal flow going through it.

I have also added further discussion of the bounds on effective pressure implied by flotation and the lake running dry in the fourth paragraph of section 5.1, where I describe the obvious modification to the lumped model that would reflect these 5 bounds.

**Referee comment:** P1, L20 write: "magnitude and timing of the flood." As for hazard prevention knowing the timing is probably equally important to the magnitude.

Done

10 Referee comment: P2, L13: "directly directed" is awkward

**Done**

**Referee comment:** Eq 1a: state that the pressure dependence of the melting point is neglected

Added the following to the introduction: Equation (1a) also ignores the effect water pressure on the melting point, which affects the fraction of the dissipated heat available for melting of the conduit walls (Werder, 2014).

15 **Referee comment:** Eq 1c: I find this equation strange. For  $v_0$  (and  $v_c$ ) no separate equation  $v_0 = v_0(S)$  is added either. Thus be consistent and just write  $q(S, \Psi)$ . Similarly in all later equations.

The reason for doing this is in the form stated is that q is a variable, not a function. Specifically, there is a term  $\partial q/\partial x$ , which doesn't make sense if q is not a function not of x, meaning I'd have to write explicitly  $\partial q(X, \Psi) / \partial x$ , which gets ugoy — the inconsistency here is really in using the same letter for variable and function. I have changed this to make  $\tilde{q}$  the function of x

**and kept q as the function of S and $\Psi$ 20**

**Referee comment:** P2: would it make sense to somewhere define what a "conduit", a "channel" and a "cavity" is? For example on P7,L4 "conduit" is used signalling the use of the  $v_0$  term again. So an unexpected reader may trip there without a clear definition.

Sure, I have now elaborated on this to the effect that a conduit is a generic drainage element, while channels and cavities are really limiting cases of a conduit that have different qualitative properties. At the start of section 2.1, I now state that: 25

[...] Note that we use the term 'conduit' here to refer to a generic drainage element that can evolve dynamically to behave as an R-channel, in which dissipation and creep closure are the dominant mechanisms by which channel size changes, or as a cavity, in which opening due to sliding over bedrock and creep closure are dominant (Kessler and Anderson, 2004; Schoof, 2010).

My mistake was probably being careless about the use of the word "channel" (e.g. p. 4, line 11) — I believe I have now 30 consistently used the word "conduit" except where I really mean "channel" in the sense of an R-channel-type balance of opening and closing terms.

**Referee comment:** *P4*, *L5*: write "background hydraulic potential" done

**Referee comment:** P4, L24: "large lakes" 35

done

**Referee comment:** *P4*, *L28: This paragraph confuses me. Is this not obvious? If not, be explicit what is odd. If it is obvious, delete.*

Deleted

5

**Referee comment:** *P5*, *L1*: "in general"

this went out of the window with the paragraph in the previous query

Figure 0: A figure depicting the used geometry would be helpful.

I've added one; not sure how much it really clarifies

Referee comment: P6, L28: "model"

10 corrected

Referee Comment: P7, L21: these "reservoirs" are, e.g., moulins. Why not state this here?

There is of course the case of very large reservoirs too, which are probably not realistic but theoretically can be stabilized, so I don't want to make a one-to-one correspondence with moulins here. In any case, the relevant paragraph has disappeared as part of a shortening of the stability section advocated by the other referee.

**Referee comment:** P8 (on this page line numbers are messed up), L5+2: q(S, N)

indeed; the relevant section has been changed significantly but I have corrected the mistake.

**Referee comment:** P8, first un-numbered Eq: this should be  $v_o$  not  $v_0$ . Or is  $v_0 = v_o(S)$ ? If so state.

This is a simple typo, corrected.

Referee comment: P9, L24-29: this describes again a moulin

I am happy to mention moulins here, subject to the same caveat as above. The updated passage reads:

The second term in (11) is a stabilizing term that is inversely related to storage capacity. Its physical origin is the following: the sensitivity of conduit growth to perturbations in conduit size may be positive, potentially leading to unstable conduit growth. However, growth of the conduit also allows water to drain out of the system, which will increase the effective pressure N. As the effective pressure is increased, the hydraulic gradient  $\Psi = \Psi_0 - N/L$  is reduced. This leads to less turbulent dissipation

25 in the conduit as the conduit grows, and increased N further leads to faster creep closure of the conduit. Both of these will suppress further growth of the conduit. How strong this stabilizing effect is will depend on the storage capacity of the system, and on the length L of the flow path: a large storage capacity  $\bar{V}_p$  or a long flow path L leads to a reduced stabilizing effect. In practice, we are likely to see stabilization.

textbfReferee comment: Fig 1: replace "lake" with "reservoir"

30 done

**Referee comment:** *Fig 1: parenthesis missing after "3.3"* corrected

**Referee comment:** *P11, L11: again* vo corrected

**Referee comment:** *P12*, *L22–23: the "immediately" needs to be weakened here. According to fig. 3: lake is empty and starts filling at the point (70,10), then the lake is filling again but S still drops from 10m2 to 0.*

This is true, but the it takes a very short amount of *time* (rather than distance in phase space) for that gap in phase space to be covered. I have changes thus to "almost immediately"

5 **Referee comment:** *Fig.3: Split the second sentence at the "and"*

Good idea, done

Referee comment: Fig.4: could the plotted Vp be added as horizontal lines to Fig. 6?

No doubt they can.

Referee comment: Fig.4: I don't understand what the line style "solid dashed" is supposed to be. I think the unstable

10 periodic should be described as "dotted coloured"

Indeed. I have no idea what I was thinking. "Dotted coloured" it is.

Referee comment: Fig.4: "insets"

corrected

**Referee comment:** Fig.5: zoom to relevant  $q_{in}$  values

15 Done

**Referee comment:** *Fig.6: it is not clear to me what is meant with "as well as in a small strip to the right of the right-hand branch of the red curve." nor is this intriguing strip ever mentioned in the text. Clarify. Maybe a zoomed inset?*

I have added the following clarifying text:

[...] as well as in a small strip to the right of the right-hand branch of the red curve: this is the region where the stable limit

20 cycles shown for instance in the inset of figure 5(b) coexist with stable steady state solutions, and exist for larger  $q_{in}$  thn the periodic orbit that reaches a minimum of N = 0 near the upper critical value of  $q_{in}$ .

Because the red curve is the locus in parameter space of periodic solutions that just reach N = 0 (rather than the boundary of the region in which N < 0 is attained by a limit cycle, which would be harder to compute), a zoom won't show anything here.

**Referee comment:** Eq 16: should be " $\sim$ "

corrected

Referee comment: P19, L16: "the its", delete "the"

corrected

Referee comment: Fig 7: "(b), 10 km"

30 corrected

35

Referee comment: Fig 7: "plotted in black" needs to be more specific. "plotted as black line"

corrected

**Referee comment:** Fig. 8/9, P32, L5: In view of this model behaviour and the potentially unstable numerics, some words should be said about the employed numerical methods (spatial discretisation, time-stepper). Yes, the code is provided but this should be in the text.

I have added the following to the end of the first paragraph of section 4:

We adapt that method for the transient nonlinear calculations in figures 9–11 by solving equations (B1)–(B2) of the same

appendix in Schoof et al (2014) using a backward Euler step.

**Referee comment:** *Fig 9: What is*  $S_{\mathcal{R}}$  *?*

5 It's an idiotic legacy notation error on my part; it should say S.

**Referee comment:** P23, L8: I don't see: "amplitude slowly grows" in fig 10

Ok, "slowly" may be an overstatement. Changed to ... first grows and then saturates

Referee comment: P25, L1: "reservoir" instead of "lake"

thanks for spotting this, corrected

10 **Referee comment:** *P26: twice wrong reference to 2012 instead of 2014* corrected

Referee comment: P26, L6: "results", P26, L21: "a spatial"

"results" has been corrected. "to have spatial structure" seems ok sans the article.

**Referee comment:** *P27, L11:*  $\Re$  *needs to be defined*

15 This has gone as part of a rewrite of Sec 5.2

Referee comment: P30, L5–9: remove if Sec 5.2 is removed

see above, I've kept this since I merely shortened section 5.2

**Referee comment:** Supplement: Excellent, that the code is published! Two things: (1) There should be a README in each zip file, stating at least which script needs to be run to produce which figures. (2) I would suggest to add a licence to each zip-file

20 (preferably an approved open source licence, the BSD-licence is popular with Matlab files https://opensource.org/licenses/BSD-3-Clause). Then it is clear under which conditions the code can be used.

I have added README files to both directories. Note that the code for the transient calculations is actually identical to that used in Rada and Schoof (2018) and the code availability statement now refers to that paper.

[revised manuscript text omitted]

$$\Psi = \Psi_0 - N/L. \tag{7c}$$

If  $q_{in} > 0$ , it follows that  $\overline{\Psi} > 0$ , and from and the properties of  $v_c$ , we also have  $\overline{N} > 0$ . For future convenience, we also write  $q(\overline{S}.\overline{N}) = \overline{q}$ . Note that, for the specific choices in , a steady state exists for every positive  $q_{in}$ . Eliminating  $\overline{S}$  and  $\overline{\Psi}$ , we find a problem for  $\overline{N}$  alone:

$$\frac{c_1 q_{in} \left(\Psi_0 - \frac{\bar{N}}{L}\right) + v_0 - c_2 \left(\frac{q_{in}}{c_3(\Psi_0 - \bar{N}/L)}\right)^{1/\alpha} \bar{N}^n = 0$$

The left-hand side is a monotonically decreasing function of  $\bar{N}$  for  $0 \le \bar{N} < \Psi_0 L$ , positive at  $\bar{N} = 0$  and tending to  $-\infty$  as  $\bar{N} \to \Psi_0 L$ , so the equation always admits a unique solution for  $\bar{N}$  in this range, from which  $\bar{S}$  can then be determined. The solution then has where we assume positive lake inflow  $q_{in} > 0$ . It can be shown (see section 3 of the supplementary material) that there is a unique steady state solution with positive  $\bar{N}$ ,  $\bar{S}$  and  $\bar{\Psi}$ .

The point of the stability analysis is ultimately to establish conditions under whether this steady state is stableand can therefore persist over time: if so, no outburst floods need to result from the presence of the reservoir. Linearizing about To establish whether the solution is stable, we can linearize around the steady state as

$$N = \overline{N} + N' \underbrace{\exp(\lambda t)}_{\swarrow}, \qquad S = \overline{S} + S' \underbrace{\exp(\lambda t)}_{\swarrow}.$$

and putting  $\bar{V}_p = V_p(\bar{N})$  gives the following leading-order form of :

$$\underline{\dot{S}'} = \underline{c_1 q_S \bar{\Psi} S' - (q_{\Psi} \bar{\Psi} + \bar{q}) L^{-1} N'} \\ + \underline{v_{o,S} S' - v_{c,S} S' - v_{c,N} N'} \\ - \bar{V}_p \dot{N}' = \underline{-q_S S' + q_{\Psi} L^{-1} N'}$$

5 This yields the following eigenvalue problem

$$\begin{pmatrix} c_{1}q_{S}\bar{\Psi} + v_{o,S} - v_{c,S} - \lambda & -c_{1}(q_{\Psi}\bar{\Psi} + \bar{q})L^{-1} - v_{c,N} \\ \bar{V}_{p}^{-1}q_{S} & -\bar{V}_{p}^{-1}q_{\Psi}L^{-1} - \lambda \end{pmatrix} \begin{pmatrix} S' \\ N' \end{pmatrix} = 0,$$
(8)

where

 $v_{c,S} =$

$$q_{S} = \left. \frac{\partial q}{\partial S} \right|_{S=\bar{S},\Psi=\bar{\Psi}}, \quad q_{\Psi} = \left. \frac{\partial q}{\partial \Psi} \right|_{S=\bar{S},\Psi=\bar{\Psi}}, \quad v_{o,S} = \left. \frac{\mathrm{d}v_{o}}{\mathrm{d}S} \right|_{S=\bar{S}},$$
$$\left. \frac{\partial v_{c}}{\partial S} \right|_{S=\bar{S},N=\bar{N}}, \quad v_{c,N} = \left. \frac{\partial v_{c}}{\partial N} \right|_{S=\bar{S},N=\bar{N}}.$$

As before, we want to know whether S' grows over time. Looking for solutions of the form  $S' = S'_0 \exp(\lambda t)$ ,  $N' = N'_0 \exp(\lambda t)$ , we get the eigenvalue problem

$$\underbrace{v_{c,S}}_{\underbrace{\longrightarrow}} = \underbrace{\mathbf{0}}_{\underbrace{\partial S}} \left| \underbrace{s_{=\bar{S},N=\bar{N}}}_{\underbrace{\otimes=\bar{S},N=\bar{N}}, } \quad \underbrace{v_{c,N}}_{\underbrace{\longrightarrow}} = \frac{\partial v_c}{\partial \underline{N}} \right| \underbrace{s_{=\bar{S},N=\bar{N}}}_{\underbrace{\otimes=\bar{S},N=\bar{N}, }} \quad \bar{V}_{\underbrace{p}=V_p(\bar{N})}_{\underbrace{\longrightarrow}}.$$

Setting the determinant of the matrix on the left<del>to zero leads to a polynomial for  $\lambda_{\tau}$ , we obtain a quadratic with solution</del>

10
$$\lambda = \frac{1}{2} \left[ a_1 \underline{\lambda + a_2 = 0} \pm \sqrt{a_1^2 - 4a_2} \right]$$
 (9)

where the coefficients take the form

$$a_1 = c_1 q_S \bar{\Psi} + v_{o,S} - v_{c,S} - \bar{V}_p^{-1} q_\Psi L^{-1}$$
(10a)

$$a_{2} = \bar{V}_{p}^{-1} \underline{q_{S}} \underline{c_{1}(q_{\Psi} + )} \underline{L^{-1} + v_{c,N}} \underline{-}_{p}^{-1} q_{\Psi} \underline{L^{-1}} \underline{c_{1}q_{S}} \underline{+} \underline{v_{o,S} - v_{c,S}} \underline{=}_{p}^{-1} \left[ c_{1}q_{S}\bar{q}L^{-1} + q_{S}v_{c,N} + q_{\Psi}L^{-1}(v_{c,S} - v_{o,S}) \right]$$
(10b)

But, from From our assumptions on the various functions involved, we see that  $a_2 > 0$  (recall that  $v_{o,S} \le 0$  from (6)), while  $a_1$ 15 can be either sign. The characteristic quadratic has solutions-

$$\lambda = \frac{1}{2} \left[ a_1 \pm \sqrt{a_1^2 - 4a_2} \right]$$

5

10

15

Since  $a_2 > 0$ , we have  $a_1^2 - 4a_2 < a_1^2$  and two possible types of solution: either  $a_1^2 - 4a_2 > 0$  and we have two real roots, both of which have the same sign as  $a_1$ . Alternatively, we have  $a_1^2 - 4a_2 < 0$  and a complex conjugate pair of roots, both of which have real part  $a_1$ . In either case, we see that the system is linearly unstable if and only if  $a_1 > 0$ , or

$$\left(c_1 q_S \bar{\Psi} + v_{o,S} - v_{c,S}\right) - \bar{V}_p^{-1} q_\Psi L^{-1} > 0.$$
(11)

We have deliberately written the left-hand side of (11) as the difference of two terms, a potentially destabilizing term  $c_1q_S\bar{\Psi} + v_{o,S} - v_{c,S}$  and a stabilizing term  $-\bar{V}_p^{-1}q_{\Psi}L^{-1}$ . The first term can be recognized as the growth rate sensitivity we previously identified as being at the heart of Nye's instability in (5), as well as an indicator of whether the steady-state conduit is 'channel-like' in the terminology of Schoof (2010).

The second term in (11) is a stabilizing term that is inversely related to storage capacity. Its physical origin is the following: the sensitivity of channel-conduit growth to perturbations in conduit size may be positive, potentially leading to unstable conduit growth. However, growth of the conduit also allows water to drain out of the system, which will increase the effective pressure N. As the effective pressure is increased, the hydraulic gradient  $\Psi = \Psi_0 - N/L$  is reduced. This leads to less turbulent dissipation in the conduit as the conduit grows, and increased N further leads to faster creep closure of the channelconduit.

Both of these will suppress further growth of the conduit.

How strong this stabilizing effect is will depend on the storage capacity of the system. If the storage capacity is large, then the growth of the channel will have a minimal impact on effective pressure (essentially, the amount of water that drains out due to the widened conduit is insufficient to affect water levels and therefore water pressure in the storage system sufficiently). The

- 20 stabilizing effect of the second term is therefore small when, and on the length L of the flow path: a large storage capacity  $\bar{V}_p$ is large. Similarly, if the system size or a long flow path L is large, then the role of effective pressure in controlling hydraulic gradients is small, and  $\Psi$  is always close to the background hydraulic gradient. The stabilizing effectis then again small, as drawing water out of the storage system does not significantly affect hydraulic gradients and turbulent dissipationleads to a reduced stabilizing effect. In practice, we are likely to see stabilization for small storage elements, such as moulins and
- 25 individual crevasses.

In summary,

**3.2 Stability boundaries**

We have established that Nye's instability will occur if the conduit is channel-like and water storage in the system is sufficiently large, and if the system length L is big enough. Note that with the choices in (1g), the system is always unstable if  $v_o = 0$  (so

30 the conduit is always a channel) and  $L = \infty$  (so effective pressure does not alter the hydraulic gradient). This is in agreement with Ng (1998).

| Parameter | value                                                                           |
|-----------|---------------------------------------------------------------------------------|
| $c_1$     | $1.3455 \times 10^{-9} \ \mathrm{J}^{-1} \ \mathrm{m}^3$                        |
| $c_2$     | $3.44 \times 10^{-24} \ \mathrm{Pa^{-3} \ s^{-1}}$                              |
| $c_3$     | $4.05 \times 10^{-2} \ \mathrm{m}^{9/4} \ \mathrm{Pa}^{-1/2} \ \mathrm{
[revised manuscript text omitted]

---

## Editor Decision (ED1)

[revised manuscript text omitted]

---

## Author Response (AR2)

**An analysis of instabilities and limit cycles in glacier-dammed reservoirs: response to reviewers**

Christian Schoof

**1 Referee # 1**

P2, L4: "same" -> "some"

ACTION: corrected

P2, L32: "behaviour typical of" ACTION: corrected

P3,L2: "It is unclear if ..."

ACTION: corrected

P3, L17: "rises such" and delete one of "alternative" and "different'"

ACTION: deleted "different"

P4, L1: "from Schoof et al."

ACTION: "as stated in Schoof et al"

P5, L10: "(Werder, 2016)", I'd also add "Röthlisberger, 1972"

ACTION: added reference

P6, L4-5: this is repeated, delete

ACTION: deleted

P6, L10: "drains"

ACTION: corrected

P7, L8: if "dN/dx" is negligible, why does it then feature in Eq.2d? State more accurately.

ACTION: added "or approximated by a simple divided difference."

P7, L20: full stop missing.

ACTION: corrected

P10, L3-10: nice!

ACTION: celebratory cup of tea

P10, L28: "N in units of m"

ACTION: corrected

25    Fig 2: the asymptote for the red line looks like it's not going to meet the line (it's slope is too shallow). Is that correct? ACTION: correct, there is an offset; the asymptote is parallel but does not meet. The derivation does not strictly hold for small L. I hope the existing text following eq (13) suffices: "where we see that they are most accurate for large L as expected"

P16, L12: "will change" -> "would change" as the presented model does not model floation.
ACTION: corrected

30    P18, L2: "drops"
ACTION: corrected

P19, L1: describe where this region is located in Fig 7 (as you do for the other asymptotic solution)
ACTION: Done ("This solution is valid towards the right-hand edge of the grey region in figure 7")

P19,L21: "effect of incipient"
35    ACTION: corrected

Fig 7: second line of caption: delete extraneous ")"
ACTION: corrected

Fig8: the reference at the end of the caption should point to figure 5.
ACTION: corrected

40    P24, L25: "based" -> "inspired" as for the Hewittal model, one needs to treat S and Sw differently. I don't think this happens in the Eq16 model. Maybe briefly mention this discrepancy.
ACTION: I think we may be going into too much detail; the box model does not have a separate sheet and channel component. I've changed the wording to "inspired"

P26, L8: "and L."
45    ACTION: corrected

P26, L10: wrong reference, I think Sec4 is right.
ACTION: corrected

P27, L27: don't cite the same paper twice in one sentence.
ACTION: corrected; the second instance should not have been there

50    P28, L24: "remains open during"
ACTION: fixed

References: please add DOIs
The copernicus bibtex class that i have doesn't seem to do that...?

[revised manuscript text omitted]